



# Resolving heterogeneous fluxes from tundra halves the growing season carbon budget

Sarah M. Ludwig[1,2], Luke Schiferl[2,3], Jacqueline Hung[4], Susan M. Natali[4], and Roisin Commane[1,2]

[1]Department of Earth and Environmental Science, Columbia University, New York, NY, United States of America
[2]Lamont-Doherty Earth Observatory, Palisades, NY, United States of America
[3]Harvard John A. Paulson School of Engineering and Applied Sciences, Cambridge, MA, United States of America
[4]Woodwell Climate Research Center, Woods Hole, MA, United States of America

**Correspondence:** Sarah M. Ludwig (ludda.ludwig@columbia.edu)

**Abstract.** Landscapes are often assumed to be homogeneous when interpreting eddy covariance fluxes, which can lead to biases when gap-filling and scaling-up observations to determine regional carbon budgets. Tundra ecosystems are heterogeneous at multiple scales, with variation in plant functional types, soil moisture, thaw depth, and microtopography, for example, influencing net ecosystem exchange (NEE) of carbon dioxide ($CO_2$) and methane ($CH_4$) fluxes. With warming temperatures, Arctic

ecosystems could change from a net sink to a net source of carbon to the atmosphere in some locations, but the carbon balance remains highly uncertain. In this study we report results from growing season NEE and $CH_4$ fluxes from an eddy covariance tower in the Yukon-Kuskokwim Delta in Alaska. We used footprint models and Bayesian Markov Chain Monte Carlo (MCMC) methods to un-mix tower observations into constituent landcover fluxes based on high resolution landcover maps of the tower region. We compared three types of footprint models and used two landcover maps with varying complexity to determine the

effects of these choices on derived ecosystem fluxes. We used artificially created gaps of withheld observations to compare gap-filling performance using our derived landcover-specific fluxes and traditional gap-filling methods that assume homogeneous landscapes. We also compared resulting regional carbon budgets when scaling-up observations using heterogeneous and homogeneous approaches. Traditional gap-filling methods performed worse at predicting artificially withheld gaps in NEE than those that accounted for heterogeneous landscapes, while there were only slight differences between footprint models and land-

cover maps. We identified and quantified hot spots of carbon fluxes in the landscape (e.g., late growing season emissions from wetlands and small ponds). We resolved distinct seasonality in tundra growing season NEE fluxes. Scaling while assuming a homogeneous landscape overestimated the growing season $CO_2$ sink by a factor of two and underestimated $CH_4$ emissions by a factor of two when compared to scaling with any method that accounts for landscape heterogeneity. We show how Bayesian MCMC, analytical footprint models, and high resolution landcover maps can be leveraged to derive detailed landcover carbon

fluxes from eddy covariance timeseries. These results demonstrate the importance of landscape heterogeneity when scaling carbon emissions across the Arctic.



# 1   Introduction

Eddy covariance towers provide some of the longest and highest resolution timeseries of in situ observations of energy, water, and carbon fluxes. Eddy covariance flux data provide landscape-level insight into numerous ecosystem processes, such as water-use efficiency, crop yields, and carbon balances (Baldocchi, 2003; Baker and Griffis, 2005; Reichstein et al., 2007; Knauer et al., 2018). Global and regional networks of eddy covariance towers, such as FLUXNET and AmeriFlux (Novick et al., 2018; Papale, 2020), are commonly used to benchmark Earth system models, provide a priori fluxes for atmospheric inversion models, or train remote-sensing based models to scale bottom-up carbon budgets (Friend et al., 2007; Wang et al., 2007; Jung et al., 2009, 2020; Chevallier et al., 2012; Tramontana et al., 2016; Chen et al., 2018; Schiferl et al., 2022). The surface source area contributing to eddy covariance flux measurements (i.e., the footprint) is much larger than other types of direct flux measurements, such as chambers, but is spatially and temporally variable, and can change with wind direction and atmospheric stability. The dynamic spatial influence on eddy covariance fluxes is often ignored under implicit assumptions that landscapes within the eddy covariance footprint are homogeneous or spatially representative (Griebel et al., 2016; Giannico et al., 2018).

Numerous footprint models have been developed to quantify source area contribution to tower flux observations (Schmid, 2002) and inform interpretations and analysis of these fluxes. Aggregate footprints are commonly used to determine the general spatial extent and seasonal patterns in tower source areas (Amiro, 1998). When combined with landcover maps of tower locations, footprints have been used to filter flux observations to include only those from distinctly uniform source areas for further interpretation and analysis, though the practicality of this is highly dependent on landscape heterogeneity and the tower site location (Jammet et al., 2017; Juutinen et al., 2022; Beckebanze et al., 2022). Studies using concurrent chamber-based fluxes within tower source areas have used footprints to scale-up chamber fluxes and compare to tower fluxes, which can provide confidence in the flux measurements, the representativeness of the chamber fluxes, and the landcover map used (Kade et al., 2012; Stoy et al., 2013; Morin et al., 2017; Davidson et al., 2017). However, disagreement between scaled chamber and tower fluxes is difficult to diagnose; chamber fluxes are often limited in temporal resolution and spatial extent, and landcover maps might not capture detail or distinctions relevant for fluxes (Fox et al., 2008; Forbrich et al., 2011; Budishchev et al., 2014). Footprints have been used to identify hotspots of methane ($CH_4$) fluxes (Matthes et al., 2014; Rößger et al., 2019; Reuss-Schmidt et al., 2019), and in circumstances where there is a single source against a known or zero flux background, the footprint-weighted flux maps can derive $CH_4$ fluxes at these hotspots (Rey-Sanchez et al., 2022). However, footprint-weighted flux maps cannot derive actual fluxes in circumstances with multiple different $CH_4$ sources, or when fluxes, such as carbon dioxide ($CO_2$), have high temporal variability. Tuovinen et al. (2019) used footprints to weight contributions to $CH_4$ fluxes from land cover classifications in heterogeneous Siberian tundra, and by assuming fluxes were constant through time, was able to solve for landcover specific $CH_4$ fluxes using ordinary least squares.

Despite the documented effects of heterogeneous surfaces on the interpretation of fluxes, most uses of eddy covariance fluxes ignore the dynamic nature of flux source areas. For applications such as model benchmarking, bottom-up scaling, and gap-filling, the landscape around towers is implicitly assumed to be homogeneous. Gap-filled timeseries are often required to create





seasonal or annual carbon budgets. One of the most widely used gap-filling approaches for $CO_2$ fluxes, marginal distribution sampling (MDS), primarily uses the mean flux of $CO_2$ from similar meteorological conditions within a certain window of time, irrespective of the wind direction and source area of the gap-filled timepoint, or of the observations used to do the filling (Reichstein et al., 2005; Wutzler et al., 2018). Both model benchmarking and bottom-up carbon flux scaling rely on tower fluxes

being spatially representative of a larger region (Williams et al., 2009). While landscape representativeness or homogeneity is a reasonable assumption for some tower sites, such as agricultural fields, it is rarely tested explicitly with footprints. A recent study by Chu et al. (2021) tested the spatial representativeness of AmeriFlux sites using footprint climatologies, and found a minority of sites were representative of areas more than one kilometer away from the tower. A recent synthesis of circumpolar $CH_4$ fluxes excluded eddy covariance measurements because they could not be unmixed and attributed to specific wetland or

waterbody sources (Kuhn et al., 2021).

Arctic ecosystems in particular require a representative network of carbon flux observations to accurately derive seasonal and annual budgets. Rapid arctic warming is thawing and mobilizing carbon stored in permafrost, leading to direct climate feedbacks through decomposition and indirect consequences through changing hydrology, vegetation, and disturbances (Rantanen et al., 2022). There is large uncertainty in the arctic carbon budget, and it remains unclear whether the arctic is currently a

carbon source or sink (McGuire et al., 2009, 2018; Natali et al., 2019, 2021; Virkkala et al., 2021; Watts et al., 2021, 2023). Tundra ecosystems are extremely heterogeneous at multiple scales (Virtanen and Ek, 2014), which when combined with logistical difficulties in monitoring in the arctic, can lead to difficulties in representative bottom-up carbon scaling (Goodrich et al., 2016; Lara et al., 2020; Pallandt et al., 2022). For example, bottom-up scaling models estimate twice as much $CH_4$ from the arctic as top-down atmospheric inversions (Thornton et al., 2016; Saunois et al., 2020).

This study addresses how landscape heterogeneity affects gap-filling and bottom-up scaling of $CO_2$ and $CH_4$ eddy covariance fluxes. We used footprint models and landcover maps to unmix eddy covariance fluxes into constituent landcover fluxes in heterogeneous tundra in the Yukon-Kuskokwim (YK) Delta, Alaska. We investigated how the choice of footprint model affects gap-filling and carbon budgets by comparing results using three of the most commonly used footprint models. We compared net ecosystem exchange (NEE) results from $CO_2$ fluxes using both a simple and a complex landcover map to determine how the

scale of heterogeneity that we consider impacts our resulting carbon budgets. Lastly, we compared gap-filled NEE fluxes and scaled-up carbon budgets to an identical approach that only differs by assuming a homogenous landscape, and to a commonly used gap-filling approach (MDS), which implicitly assumes a homogeneous landscape. We discuss the implications of the resulting $CO_2$ and $CH_4$ fluxes and carbon budgets for the YK Delta and arctic carbon feedbacks.

## 2 Methods

### 2.1 Site Description

The study region is located in the Izaviknek and Kingaglia Uplands of the Yukon-Kuskokwim Delta in Alaska, approximately 90 km northwest of Bethel, Alaska and 110 km inland from the coast. Mean annual air temperature in Bethel was 1.2 °C for 2019-2020, 13.8 degrees C during summer (June, July, August), -11.9 °C during winter (December, January, and February),





and above freezing from May-October. The study region is underlain by discontinuous permafrost, with permafrost underlying

peat plateaus and absent under wetlands and lakes (Frost et al., 2020). Thaw depths on peat plateaus averaged 30-40 cm in June and July 2016-2017 and 60-70 cm in September 2016 (Ludwig et al., 2022). Vegetation on the peat plateaus is heterogeneous, and is dominated by lichen (primarily *Cladonia* spp), *Sphagnum fuscum*, or low-lying shrubs, while wetland vegetation is typically *Sphagnum* and graminoid spp. (Zolkos et al., 2022). The eddy covariance tower was installed in July 2019 on a peat plateau in unburned tundra at (N 61.2548°, W 163.2589°).

95        We used a landcover map developed by Ludwig et al. (2022) to characterize the eddy tower location and a nearby region of unburned tundra used for scaling up carbon budget (Ludwig et al., 2023a). The landcover map is 5 m x 10 m resolution and derived from Sentinel-1 synthetic aperture radar (SAR), Sentinel-2 multispectral instrument (MSI), and the ArcticDEM. Two versions of landcover were used: (1) a simple version with only four categories: surface water, tundra, wetland, and degrading permafrost, and (2) a complex version where tundra was further split into lichen tundra, shrub tundra, sedge tundra,

and tundra at the edge of degrading permafrost (Fig 1). Tundra landcover categories were primarily located on peat plateaus, and share the same dominant vegetation types of lichens, dwarf shrubs, mosses, and sedges. The differences within tundra categories were subtle; shrub tundra was often located at the edges of peat plateaus bordering and along banks with slightly larger shrubs; sedge tundra was located on peat plateau slopes that were slightly greener; lichen tundra was the least green and largest area of tundra types within the region, dominated by lichen, moss (*Sphagnum* spp. and *Dicranum* spp.), graminoids

(*Carex* spp. and *Eriophorum angustifolium*) (Baillargeon et al., 2022); and edge of degraded tundra included tundra bordering degraded permafrost, often wetter, mossier, and slightly subsided. Degraded areas included isolated shallow depressions on peat plateaus, more evolved networks of flowpaths draining peat plateaus into wetlands, and recently drained waterbodies. Depending on seasonality and antecedent rain, degraded areas could have standing water, saturated soils, exposed mud, or graminoid-dominated vegetation. The wetland category included a range of wetland vegetation such as mosses, graminoids,

and tall shrubs, often with complex underlying hydrology. Wetland soils were usually saturated, with small, sub-pixel channels or waterbodies undetectable at the resolution of the landcover map. Surface water includes all lakes, ponds, and streams detectable at the landcover map resolution (Ludwig et al., 2023b). There are likely smaller ponds or channels within wetlands and degraded areas, but higher resolution mapping would be needed to identify that level of heterogeneity. The full distribution of landcover areas in a 300 m radius circle around the tower location and in the region used for scaling is described in table

1. The scaling region was approximately 150 km2, which is similar to the average size of a grid cell in earth system models (Williams et al., 2009).

## 2.2   Eddy covariance data processing

Data used in this study span from July 12th 2019 to September 30th 2020, though we only include May through September months. The tower instrumentation consisted of a Gill WindMaster Pro sonic anemometer, LI-7500DS open path analyzer for

$CO_2$ and $H_2O$, LI-7700 for $CH_4$, Vaisala HMP155 humidity and temperature probe, LI-190R quantum sensor for photosynthet-ically active radiation (PAR), Kipp and Zonen CNR4 four component net radiometer, and HukseFlux HFP01SC soil heat plates. All instrumentation was connected to a LI-7550 interface equipped with a LICOR SmartFlux system. The measurement height



**Table 1.** Landcover category percentages in the immediate eddy covariance tower area (radius of 300 meters) and in the region used to scale-up ecosystem carbon fluxes (Fig 1).

| Landcover category | Tower area | Scaling region |
|---|---|---|
| Lichen tundra | 12 % | 27 % |
| Shrub tundra | 18 % | 11 % |
| Sedge tundra | 23 % | 16 % |
| Edge of degraded permafrost | 21 % | 11 % |
| Degraded permafrost | 3 % | 5 % |
| Wetland | 20 % | 18 % |
| Water | 3 % | 12 % |

was 2.5 meters above ground level. Half-hourly flux calculations were made using the eddy covariance method (Baldocchi et al., 1988) from 10 Hz data using the EddyPro software program (Fratini and Mauder, 2014). We used the double coordinate
rotation method, spike removal, block averaging, and time lag removal by covariance maximization (Moncrieff et al., 1997). We made corrections for air density fluctuations for $CO_2$, $CH_4$, and H2O fluxes following (Webb et al. 1980). Fluxes with nonstationarity were removed (Foken et al., 2004). Fluxes were further filtered to remove times of low signal strength (rssi < 15%) and low turbulence (u* < 0.1 threshold was chosen where $CO_2$ fluxes were independent of u*). Energy balance closure at the site was good (70%). Lastly, fluxes were filtered to remove spikes using the double median absolute deviation method
(Mauder et al., 2013). The resulting timeseries had 26% and 61% missing data for $CO_2$ and $CH_4$ fluxes respectively. Due to limited access for site maintenance during the COVID-19 pandemic and the remote site location, power outages contributed to 1.5% missing data in fluxes, air temperature, and PAR. While only actual observations of air temperature and PAR were used for training gap-filling models, we used a complete timeseries of drivers for scaling and to sum fluxes to monthly carbon budgets. To interpolate missing data in air temperature and PAR for scaling we used the marginal distribution sampling and mean
diurnal course method from REddyProc (Wutzler et al., 2018). Annual timeseries of $CO_2$ fluxes, $CH_4$ fluxes, air temperature, and PAR observations can be found in the SI (Fig S1a-d).

## 2.3 Eddy covariance data processing

We compared three commonly used footprint models to determine source areas for fluxes: the Hsieh model (with the 2D extension from Detto et al. (2006)), the Kljun model, and the Kormann and Meixner model (Hsieh et al., 2000; Kormann and
Meixner, 2001; Kljun et al., 2015). The Hsieh model is a hybrid approach blending a forward Lagrangian stochastic numerical model with an analytical solution. The Kljun model uses multiple parameterizations of a backward Lagrangian particle model to be applicable across atmospheric stability regimes. The Kormann and Meixner model is a Eulerian analytical footprint model based on Monin-Obukhov similarity theory. All three footprint models assume Gaussian dispersion in the crosswind direction and horizontal homogeneity in turbulence effects (Schmid, 2002). Given the flat deltaic landscape and extremely short tundra





**Figure 1.** Scaling region within the YK Delta used in this study. Sentinel-2 RGB imagery (a) with the location of the grid within Alaska as an inset, simple landcover map (b), complex landcover map (c).

canopy height relative to instrument measurement height, this site was an ideal location for footprint modeling, while still encompassing heterogeneity in $CO_2$ and $CH_4$ fluxes. We calculated a single roughness length for the site (0.02 m) from the measured wind speed and friction velocity under neutral conditions assuming a logarithmic wind profile. For each half-hour flux observation, 1 x 1 meter grid footprints were generated using each of the three model types, and then rotated into the wind direction. These footprints were then reprojected to match the resolution and extent of the landcover maps at 5 x 10 meters.

Footprints were normalized to total 100% by dividing by the sum of the weights within each observation. The footprint weights were then summed over each landcover type ($k$) for each flux observation ($i$) as $\Omega_{i,k}$ (see equation 1 in section 2.4.1).





### 2.4 Gap-filling models

We compared several approaches for gap-filling the eddy covariance NEE timeseries. First, we explicitly consider landscape heterogeneity by unmixing tower fluxes using each of the three types of footprint models when summarized over both the simple and complex landcover map (Section 2.4.1). In order to do so, NEE fluxes were partitioned into respiration and gross primary productivity (GPP) with simple empirical models driven by PAR and air temperature. Second, we used the same method of flux partitioning, modeling, and parameter estimation to gap-fill NEE, but instead assume a homogeneous landscape. Each of the heterogeneous types of gap-filling models and the similar homogeneous variation were trained separately for each month in the growing season (May through September) to accommodate seasonality. Observations from both 2019 and 2020 were used to train the gap-filling models, though we only predicted and scaled for 2020, since the 2019 growing season was incomplete. We tested the inclusion of 2019 observations for August and September, and there was little effect on the derived landcover fluxes. Last, we compare these results to a widely-used approach by gap-filling NEE with MDS, which implicitly assumes a homogeneous landscape. $CH_4$ fluxes were not as temporally variable as NEE and largely unrelated to biometeorological drivers measured at the tower. $CH_4$ fluxes were subsequently treated as landcover-specific constant fluxes through time and solved for separately in each month of the growing season.

### 2.4.1 Heterogeneous gap-filling models

Assuming that every pixel within a landcover type is characterized by a similar flux, then for a given ($k^{th}$) half-hour measurement, the observed tower NEE flux is the sum of each ($i^{th}$) landcover flux ($NEE_{i,k}$) times the total influence of those pixels within a footprint ($\Omega_{i,k}$) across all ($P$) landcovers equation 1.

$$NEE_{Obs,k} = \sum_{i=1}^{P} NEE_{i,k} * \Omega_{i,k} \tag{1}$$

If the landcover-specific NEE fluxes were constant in time, then they could be solved for using ordinary least squares, such as Tuovinen et al. (2019) do for $CH_4$ fluxes. However, $CO_2$ fluxes are often highly variable in time, especially from vegetated environments. Tundra NEE has been well characterized as the difference between respiration—modeled as an exponential function of temperature—and gross primary productivity—modeled as a light-saturating response curve often attenuated by temperature or vapor pressure deficit (Williams et al., 2006; Shaver et al., 2007; Loranty et al., 2011). For the heterogeneous gap-filling models, we structured the ($NEE_{i,k}$) fluxes from vegetated landcovers as temporally variable and dependent on air temperature ($Tair_k$), light ($PAR_k$), and air temperature rescaled between 0 and 1 ($Tscale_k$):

$$NEE_{i,k} = R_{i,k} - GPP_{i,k} \tag{2}$$

$$R_{i,k} = \alpha_i * e^{\beta_i * Tair_k} \tag{3}$$

$$GPP_{i,k} = Tscale_k * \frac{E0_i * Pmax_i * PAR_k}{Pmax_i + E0_i * PAR_k} \tag{4}$$



The parameters are $\alpha_i$ the baseline respiration, $\beta_i$ the temperature sensitivity of respiration, $E0_i$ the light-use efficiency of GPP, and $Pmax_i$ the maximum photosynthetic capacity. For the simple landcover map, NEE fluxes from tundra, wetland and

degrading permafrost were all parameterized according to equations 2-4. Surface water $CO_2$ fluxes were parameterized as a constant flux over time. While this is likely an over-simplification, a more complex lake emissions model was not feasible because the surface waters within the footprint were too small an area and too small in footprint influence to inform a more complex model. Similarly, for the complex landcover map, all NEE fluxes from tundra landcover types as well as from wetland and degrading permafrost were structured according to equations 2-4 with water as a constant flux.

An alternative model structure for GPP was investigated that uses leaf area index (LAI) as a driver (Shaver et al., 2007). In lieu of field-based LAI data, we used a timeseries of NDVI from cloud-free Sentinel-2 imagery and the empirical relationship to LAI from pan-Arctic tundra described in Shaver et al. (2013). The LAI-version GPP model failed posterior predictive checks for most months of data, and was not further pursued. This failure is likely because the approximation from NDVI was a poor representation of LAI for this site, particularly during May, August, and September where sub-pixel water presence could lead

to erroneous NDVI and LAI. Furthermore, lichen and moss species dominated the vegetation biomass on peat plateaus and LAI may not be an appropriate metric in such cases. However, a spatially resolved driver such as LAI might be effective in other applications for un-mixing NEE, particularly if LAI can be mapped from higher resolution imagery based on site-specific field-observations.

$CH_4$ fluxes were assumed to be constant over time for each landcover type.

$$CH_{4,Obs,k} = \sum_{i=1}^{P} CH_{4,i} * \Omega_{i,k} \tag{5}$$

The only parameters in this simpler version of unmixing are the landcover $CH_4$ fluxes themselves, $CH_{4,i}$ with the footprint influences $(\Omega_{i,k})$ as the only time-variable driver equation 5. All three footprint models were similarly compared for $CH_4$ fluxes. Only the complex landcover map was used to unmix $CH_4$ fluxes, since the categories within tundra were known to be divergent, e.g., known very small fluxes from lichen tundra, while tundra at the edge of degraded could possibly be a large

source.

### 2.4.2 Parameter estimation and flux prediction

We unmixed tower fluxes to landcover $CH_4$ and $NEE_{i,k}$ fluxes by using a Bayesian analysis with Markov Chain Monte Carlo (MCMC) simulation. We chose this method partly because unmixing approaches such as ordinary least squares (Tuovinen et al., 2019) are not applicable with the non-linear relationships used here between $CO_2$ and air temperature and PAR. In

addition, there are several advantages to using a Bayesian approach to solve for landcover fluxes. First, we can provide prior information on flux parameters. This prior information could be specific (e.g., from chamber fluxes from landcovers within the footprints), it could be more general (e.g., dictating one landcover known to have higher GPP than another), or it could be mostly uninformative, and merely place restrictions on parameter space based on physical properties (e.g., non-negative $Pmax_i$). We used the latter approach to NEE priors for this study to be comparable between footprint model and landcover



map solutions, and to better demonstrate the impacts of unmixing eddy tower NEE on gap-filling accuracy and bottom-up scaling. Given the simpler approach used to unmix $CH_4$ fluxes, there were multiple solutions if all prior fluxes were strictly uninformative. We used mostly uninformative prior fluxes for landcovers anticipated to support $CH_4$ emissions by disallowing $CH_4$ uptake for degraded, edge of degraded, wetland, and water landcover classes. Peat plateau chamber flux measurements from 2017 demonstrate a very small but non-zero $CH_4$ flux at the driest time of the growing season (Ludwig et al., 2018), and

we assigned prior fluxes for tundra types accordingly. Prior distributions can be found in the SI (Table S1-S3). The second benefit of using Bayesian analysis with MCMC is that derived quantities and predictions of new data are inherently treated as random variables with their own probability distributions, thus enabling easy calculations of uncertainties. Therefore, we carry through uncertainty from both partitioning and gap-filling to uncertainty in predicted landcover $NEE_{i,k}$ or $CH_4$ fluxes, which, when summed over time and scaled up by area, leads to distributions of carbon budgets from which we can calculate explicit

uncertainties.

    For each month of the growing season (May-September), gap-filling models were fit separately for each footprint-type and landcover map combination. First, $NEE_{Obs,k}$ were filtered to dark data (PAR < 50 ppfd) and respiration parameters (equation 3) were determined while using uninformative priors. The GPP parameters (equation 4) were then estimated using the $NEE_{Obs,k}$ from the full dataset, with uninformative GPP priors but using the posterior distributions of the respiration

parameters as strict prior information for the respiration component of equation 2. $CH_4$ fluxes were fit separately by month as well, but all times of day were used. We used Gibbs sampler for the MCMC iterations (Just Another Gibbs Sampler; JAGS) implemented with the runjags R package (Denwood, 2016), with a burn-in of 5,000 iterations, an adaptation of 5,000 iterations, and retained 3,000 iterations in the final chains. Three parallel chains were used for each model with different initial parameter values (Table S1-3). We evaluated parameter convergence using the Gelman diagnostic (Gelman and Rubin, 1992; Brooks and

Gelman, 1998). Model performance was further checked using posterior predictive checks of the mean, standard deviation, and sum of squared residuals (Gelman et al., 1996).

### 2.4.3  Homogeneous gap-filling models

We used several methods to gap-fill the tower NEE and $CH_4$ fluxes that both assume a homogeneous landscape/footprint for comparison. The first method is a Bayesian analysis with MCMC sampling that mirrors our landcover flux un-mixing

approach in every way except by assuming a homogeneous landscape. For the homogeneous Bayesian model, we assume a single landcover type everywhere that accounts for 100% of the footprint influence at every flux observation. The homogeneous landcover NEE was modeled monthly by equations 2-4, with the same partitioning and parameter estimation as described in section 2.4.2. The second homogeneous NEE gap-filling approach we used was MDS (Reichstein et al., 2005). We used the REddyProc package with default settings to implement the MDS gap-filling (Wutzler et al., 2018). Since the $CH_4$ fluxes did

not have relationships to any biometeorological drivers, we estimated monthly budgets by calculating the average $CH_4$ flux for each half hour period across the month, and then applying these averages to each day within the month.





### 2.4.4 Artificial gaps

Artificial gaps in the NEE and $CH_4$ flux observation timeseries were created in order to be able to evaluate and compare gap-filling approaches. Since the MDS gap-filling method requires at least 90 days of half-hourly measurements, it could only be applied to the 2020 growing season (data in 2019 began in mid-July). Therefore, we only created artificial gaps in the 2020 growing season for comparability. Artificial gaps were generated separately for each month to ensure each portion of the growing season had a similar amount of withheld data. Between 15-20% of the timeseries was withheld as random artificial gaps of stratified sizes, with $\approx$ 5% as larger gaps (10 observations), $\approx$ 5% as smaller gaps (4 observations), and the remainder as single gaps. The withheld drivers corresponding to the artificial gaps (PAR, air temperature, footprint weights) were then used to predict tower NEE or $CH_4$, and gap-filling methods were evaluated by calculating the root mean square error (RMSE).

### 2.5 Scaling up NEE and $CH_4$

We used the parameter posterior distributions from MCMC simulations and full timeseries of air temperature and PAR to predict complete, gap-filled, $CH_4$ and $NEE_{i,k}$ flux timeseries for the landcover types as described in sections 2.4.2. We then summed these distributions of half-hourly fluxes over time and multiplied by their respective areas in the scaling region (section 2.1, Fig 1) to determine estimations of monthly carbon budgets for each landcover type. Monthly landcover carbon budgets were calculated for each footprint model and landcover map combination. Landcover carbon budgets were then summed to create monthly and growing season carbon budgets for each footprint model and map type. Monthly and growing season carbon budget distributions for the Bayesian homogeneous gap-filling models were similarly estimated. The observed tower NEE fluxes with MDS gap-filling were also summed over time and multiplied by the total scaling region to arrive at comparative monthly carbon budget estimates. $CH_4$ was presented alongside NEE in carbon budgets as $CO_2$-equivalents ($CO_2$-eq) by multiplying by a factor of 28, a commonly used approximation of relative global warming potential. For discussion of uncertainty in carbon budgets, we calculate 89% Bayesian credible intervals (CI), which are analogous to frequentist 95% confidence intervals (Kruschke, 2014; McElreath, 2015; Hobbs and Hooten, 2015), using the highest density interval method from the 'baysetestR' package (Makowski et al., 2019).

## 3 Results & Discussion

### 3.1 Footprint influence

The most common and most influential landcover within the footprints was tundra, averaging 70% influence over the growing season for all three footprint model types. There was a fairly even distribution between tundra category types, though sedge tundra had slightly more influence than lichen, shrub, and degraded edges. The footprint influence from wetlands was comparable to that of individual tundra types. The least represented landcovers were degraded permafrost and surface water, both of which were between 0-20% influence (Fig S2). The vast majority of footprints were a mix of landcover types, with almost no individual footprints having a landcover type with more than 75% influence (Fig S2). There was a surprising amount of agree-



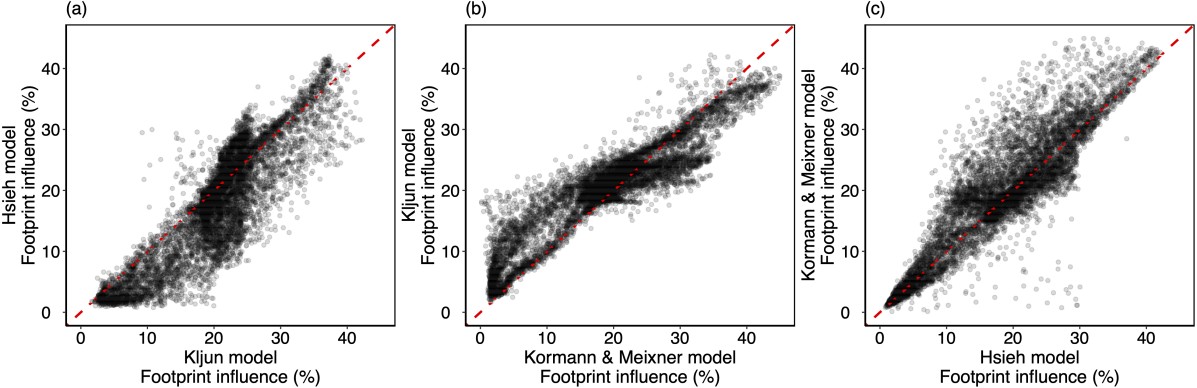

**Figure 2.** Scatterplot demonstrating comparison of footprint influence weights between the three models (Hsieh, Kljun, Kormann and Meixner) for lichen tundra. Other landcover footprint influence comparisons can be found in the SI (Fig S2). The dashed red line indicates the 1:1 line.

ment between the three footprint models, with the majority of footprint influences close to the 1:1 line on regressions between model types (Fig 2, Fig S2). Other studies that have sought to compare the ability of these footprint models to recover known

flux sources have found little distinction between them, despite the differences in their methodology (Coates et al., 2021; Rey-Sanchez et al., 2022). However, the landcover influences used here were sums of all pixel influences within a landcover type, therefore small differences between models on a pixel basis were cumulative and would lead to larger discrepancies overall. There are also distinct periods of larger differences between footprint models, likely when the peak footprint influence was near the boundary between two landcover types; thus, a small shift in peak location between model types would lead to a large

difference in landcover influences. A higher resolution landcover map (e.g., 3 m or smaller) would minimize some footprint model discrepancies, though this is relative to the extent of the footprints and the scale of landscape heterogeneity affecting carbon fluxes.

### 3.2 Model performance

All posterior predictive checks were passed (Bayesian p-values of $0.1 < p < 0.9$), and all parameters converged (Gelman

diagnostics $\approx 1$) for every Bayesian gap-filling model. All Bayesian gap-filling models were able to accurately reconstruct the tower NEE across the growing season as a function of PAR, air temperature, and source attributions. The only notable deviations were exclusive to outliers in tower NEE observations. This result is not unexpected, as eddy covariance data are often noisy. Mismatch with tower NEE outliers could also be a consequence of processes dominating fluxes that were not represented in our models, e.g., high $CO_2$ emissions from ebullition aligning with high lake influence within a footprint. When

comparing performance for filling the same artificial gaps, all Bayesian models had a better (lower) RMSE than the MDS method (Fig 3). The Bayesian models, both heterogeneous and homogeneous, drive NEE as deterministic functions of PAR and temperature. This may be why they were more accurate than MDS, which has been shown to be biased in high-latitudes due





to the effects of skewed distributions of net radiation (Vekuri et al., 2023). The heterogeneous gap-filling models almost always performed better than their homogeneous equivalent (Fig 3). For most months, the heterogeneous complex map solutions

outperformed those of the simple map (Fig 3). This was particularly notable near the shoulder seasons, where tundra types exhibited greater differences in seasonality. For example, in May the lichen tundra had very little GPP while the sedge tundra was a distinct carbon sink (Fig 4), and this could be accommodated in the complex map while the simple map attempted to fit both to a single 'tundra' flux. While there were clear improvements in gap-filling RMSE using this un-mixing method, the differences in RMSE were small relative to the magnitude of the fluxes. The drawback of the flux un-mixing method used

here are site-specific solutions and longer computation times, which increase with the landscape complexity considered. MDS remains faster to implement and could be preferred when landscape homogeneity can be safely assumed.

   None of the three footprint models consistently performed better in terms of RMSE, and for most outcomes, the distributions about their RMSEs overlapped (Fig 3, Fig S3). However, more often than not the Hsieh and Kljun footprint models performed better than the Kormann and Meixner model. Given that none of the three footprint model types quantify their uncertainty,

we continued to evaluate all three as an ensemble of footprint models that represents instead the range in footprint influence outcomes. Another way to evaluate performance of the three footprint models is by comparing their consistency in predicting landcover NEE fluxes when the underlying landcover map switches from simple to complex. The degraded permafrost, water, and wetland landcovers were identical between the two maps and ideally should have the same derived fluxes even if the tundra categories were treated differently. Similarly, the overall tundra footprint-weighted flux should match between simple

and complex landcovers, even though the complex tundra was a combination of four types where there was only one tundra type for the simple map. By weighting the predicted landcover NEE by their respective footprint influences for each observation, we regressed the simple vs. complex solutions (Fig S4-8). While the tower NEE and tundra total weighted NEE were very consistent between landcover maps for all footprint models, the Kljun footprint model was distinctly more consistent for the less represented landcovers (wetland, water, and degraded permafrost) (Fig S4-8). This outcome might indicate the Kljun

footprint model was more representative of landcover influences. In the absence of extensive concurrent chamber fluxes to conclusively distinguish between derived landcovers from the footprint models, we recommend a footprint model ensemble approach.

### 3.3 Derived landcover fluxes

There was enough similarity between footprint model influences to yield similar patterns in derived landcover fluxes (Fig 4,

Figure S9, Table S4-10). For example, in all three footprints both shrub tundra and tundra at the edge of degrading permafrost had higher peak carbon uptake than sedge and lichen tundra (Fig 4). This aligns with previous studies that have found higher productivity in shrub tundra and areas adjacent to disturbed tundra, possibly the result of increased nutrient availability (Schuur et al., 2007; Bowden et al., 2008; Lee et al., 2011). The range of NEE fluxes derived for tundra vegetation was similar to ranges in NEE observed at other tundra sites (Euskirchen et al., 2012; Howard et al., 2020; Virkkala et al., 2022). All three footprint

models also derived higher $CO_2$ and $CH_4$ emissions from surface water and wetlands later in the growing season (Fig 4, Fig 5), which could be the result of increased thaw depths contributing to greater lateral carbon transport from peat plateaus.



**Figure 3.** Monthly RMSE (a-e) and growing seasonal total (f) for 2020 from artificial gap-filling NEE fluxes. Boxes are the median and interquartile range (IQR), whiskers are 1.5*IQR, for the Bayesian model gap-filling RMSEs. The red line indicates MDS (marginal distribution sampling) gap-filling RMSEs. Note the scales on y-axes are different between panels to highlight the comparability of footprint models and landcover maps within months.

Porewater dissolved organic carbon and dissolved $CO_2$ and $CH_4$ was extremely high on peat plateaus during the growing season (Zolkos et al., 2022), and open water in both wetlands (sub-pixel) and waterbodies were likely hotspots for decomposition and


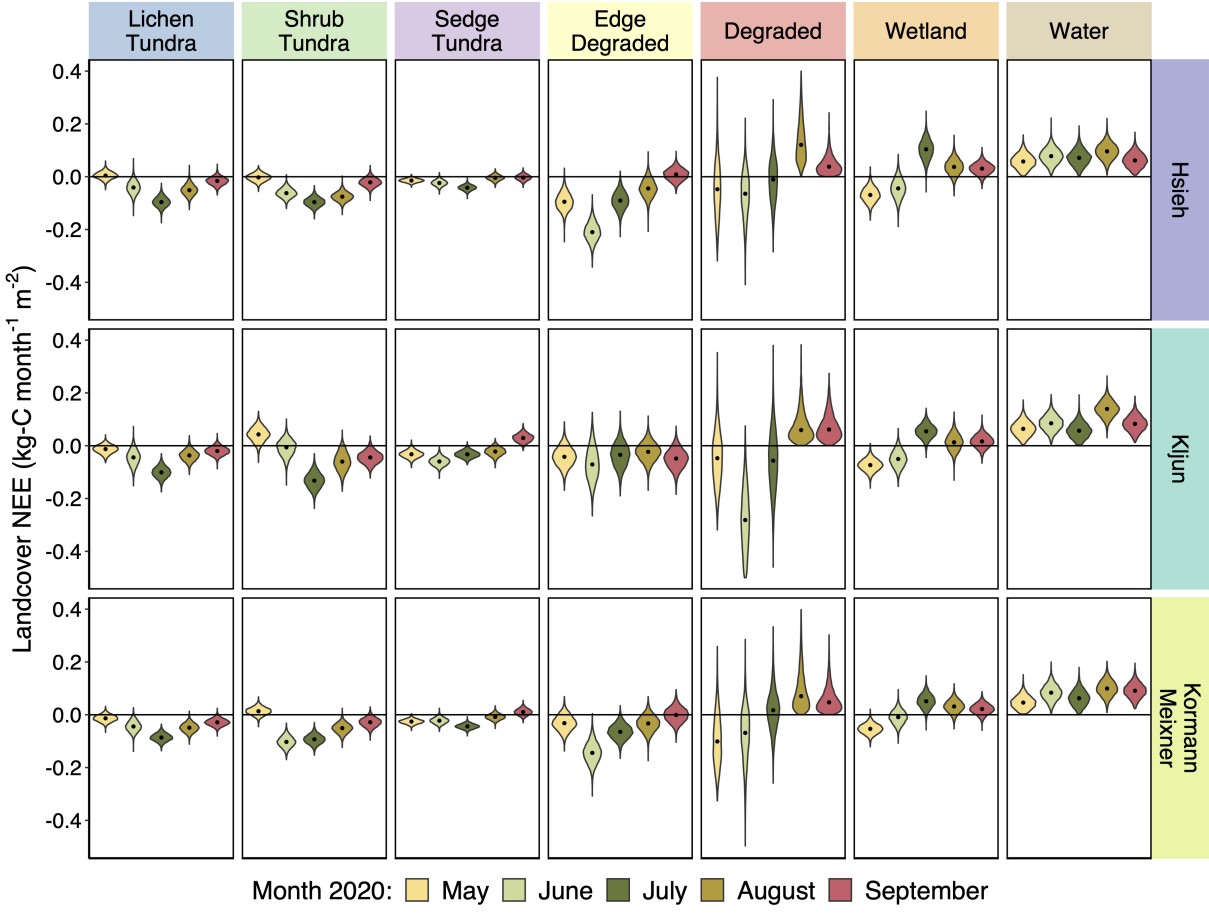

**Figure 4.** Monthly violin plots of predicted NEE fluxes from 2020 growing season by landcover (columns) for each of the three footprint models (rows) using the complex landcover map to un-mix the tower fluxes. Distributions for violin plots are derived from posterior distributions of predicted NEE. Black dots indicate medians.

outgassing (Ludwig et al., 2022). The wetlands were also characterized by deep, carbon-rich soil, which could be contributing
to higher baseline respiration (Fig 4, Table S9). The derived $CH_4$ fluxes from landcover classes in this study were within the
ranges reported in the Boreal-Arctic Wetland and Lake Dataset (BAWLD-$CH_4$), including from: wetlands and edge of degraded
permafrost ( wetlands and wet tundra in BAWLD); peat plateau (dry tundra in BAWLD); and waterbody $CH_4$ fluxes (small
peatland lakes in BAWLD) (Kuhn et al., 2021). The $CO_2$ fluxes reported here are similar in range to those observed in small
ponds in other subarctic tundra ecosystems (Kuhn et al., 2018).
The three footprint models followed similar patterns in peat plateau seasonality as well, with NEE uptake peaking in July
for most tundra types (Fig 4). Lichen and sedge tundra were very small $CH_4$ sources (Fig S9), though given the large area of
lichen tundra in the landscape this resulted in a notable contribution to total $CH_4$ (Fig 5) when scaling up. Shrub tundra was
either zero or a very small $CH_4$ source, depending on the footprint model (Fig 5). Tundra at the edge of degrading permafrost



was a significant $CH_4$ source, and behaved more similarly to wetlands than degraded areas in terms of seasonal patterns (Fig 5).
Interestingly, degraded permafrost was a sink of $CO_2$ earlier in the growing season (Fig 4), but all GPP parameters converged to zero in August and September (Fig 4, Table S8). Degraded permafrost was a source of $CH_4$ early in the growing season, decreased near to zero as the depressions dried down, and then increased again later in the growing season (Fig 5, Fig S9). This aligns with the wettest portion of the growing season, when the small depressions of degrading permafrost become inundated as small ponds (Mullen et al., 2023), which could explain the renewed $CH_4$ emissions and decline in GPP.

There were differences as well between the derived landcover fluxes for the three footprint models. These differences were off-setting between adjacent landcover types. For example, degrading permafrost and tundra at the edge of degrading permafrost were always, by definition, near one another. When the Kljun model had high carbon uptake in degrading permafrost it had lower uptake at the edge of degrading permafrost, where on the other hand, Hsieh and Kormann and Meixner displayed the opposite pattern (Fig 4). This discrepancy was the result of slight differences between footprint models in peak influence
positioning at the boundary of the two landcovers (Fig 2, Fig S2). The differences in effects of the footprint models can likely be minimized by using a relatively higher resolution map, or including spatial drivers such as LAI, soil moisture, and soil temperature, which would provide further constraints for landcover fluxes.

The complex heterogeneous models captured distinctive seasonality; lichen and shrub tundra were net neutral in May, had peak carbon uptake in July, and remained small sinks in September (Fig 4). In contrast, the sedge tundra and edge of degrading
permafrost were small sinks in May, peaked earlier, and were net neutral or carbon sources by September (Fig 4). Increasing the complexity of the underlying map allowed us to determine this separate but ecologically significant seasonality in peat plateau carbon cycling. However, there is a limit to how complex one can get. Attempting to use multiple wetland types failed, as the less prevalent wetland could not to converge, and we subsequently lumped all wetland categories as 'wetland'. This failure to converge serves as a check against over-fitting, in addition to comparing to withheld data via artificial gaps. In the event that a
landcover with small area or influence is significant to the research questions posed, but this un-mixing method cannot derive a flux, then we recommend supplying stricter prior information for the landcover via chamber fluxes. By combining chamber fluxes and eddy tower flux un-mixing, one can leverage both the spatial coverage and temporal frequency of tower fluxes with the specificity of chamber fluxes.

### 3.4 Landcover scaling

Scaling up fluxes to the region led to distinct landcover hotspots of carbon sinks and sources, with all three footprint models having similar monthly NEE and $CH_4$ budgets by landcover type (Fig 5, Fig 6). Lichen tundra was the largest sink of carbon (July carbon uptake 89% CI for Hsieh (-2628 to -5075), Kljun (-2715 to -5452), and Kormann and Meixner (-2493 to -4428) Mg-C), though this was driven in part by occupying the largest area in the region (Fig 1; Table 1). Wetlands and surface waters were significant sources of both $CO_2$ and $CH_4$ in the latter half of the growing season (July-September), with large enough
emissions to offset the carbon uptake in some of the peat plateau landcovers (Fig 5, Fig 6). Wetlands account for, on average, only 7% of what the tower sees, but 18% of the area in the region. If we were to use a coarser landcover map, such as the recently updated circumpolar arctic vegetation map (CAVM) (Raynolds and Walker, 2022), we would have attributed 100%

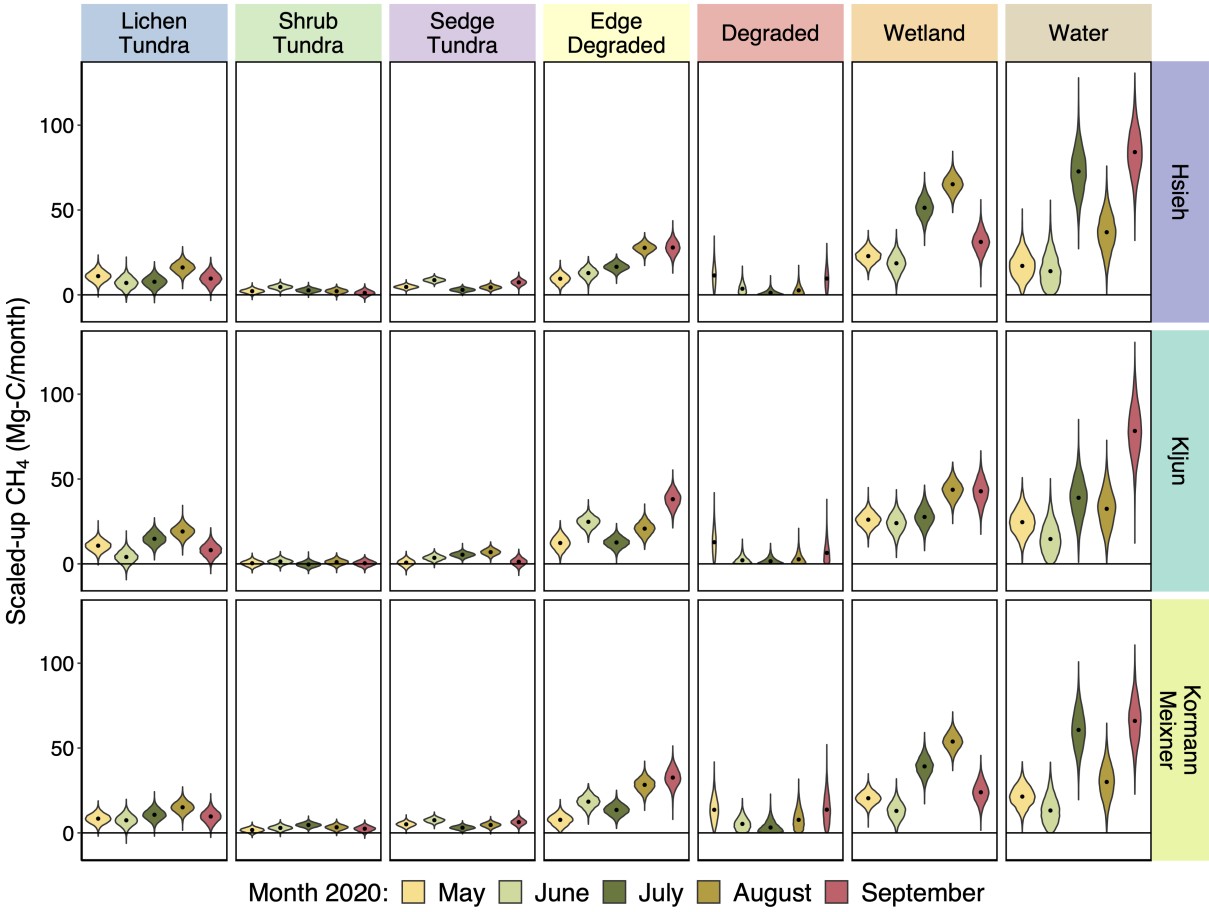

**Figure 5.** Monthly violin plots of CH$_4$ 2020 growing season budgets by landcovers (columns) for each footprint model (rows). Distributions for violin plots are derived from posterior distributions of predicted CH$_4$ fluxes scaled by their landcover areas in figure 1. Black dots indicate medians.

of the tower fluxes wetland-complex vegetation, which would scale up to 26% of the region using CAVM. Given the clearly distinctive carbon dynamics between wetlands and peat plateau vegetation, using a landcover resolution appropriate to the scale
of heterogeneity is important for obtaining an accurate regional carbon budget and understanding of the ecosystem.

Regional surface water carbon emissions scaled-up from the eddy covariance tower fluxes are likely an overestimate, since the waterbodies within the tower footprints were amongst the two smallest size-classes of waterbodies in the region, which have the largest diffusive carbon fluxes (Ludwig et al., 2023b). In comparison, the coarser CAVM landcover map does not identify any surface waters in the entire scaling region, which would lead to these hotspots of emissions being completely
underestimated. Scaling-up surface water carbon emissions would be better done using an approach that includes both terrestrial and aquatic landscape drivers and uses better spatial representation (e.g., Ludwig et al. (2023b)) than the area seen by a single





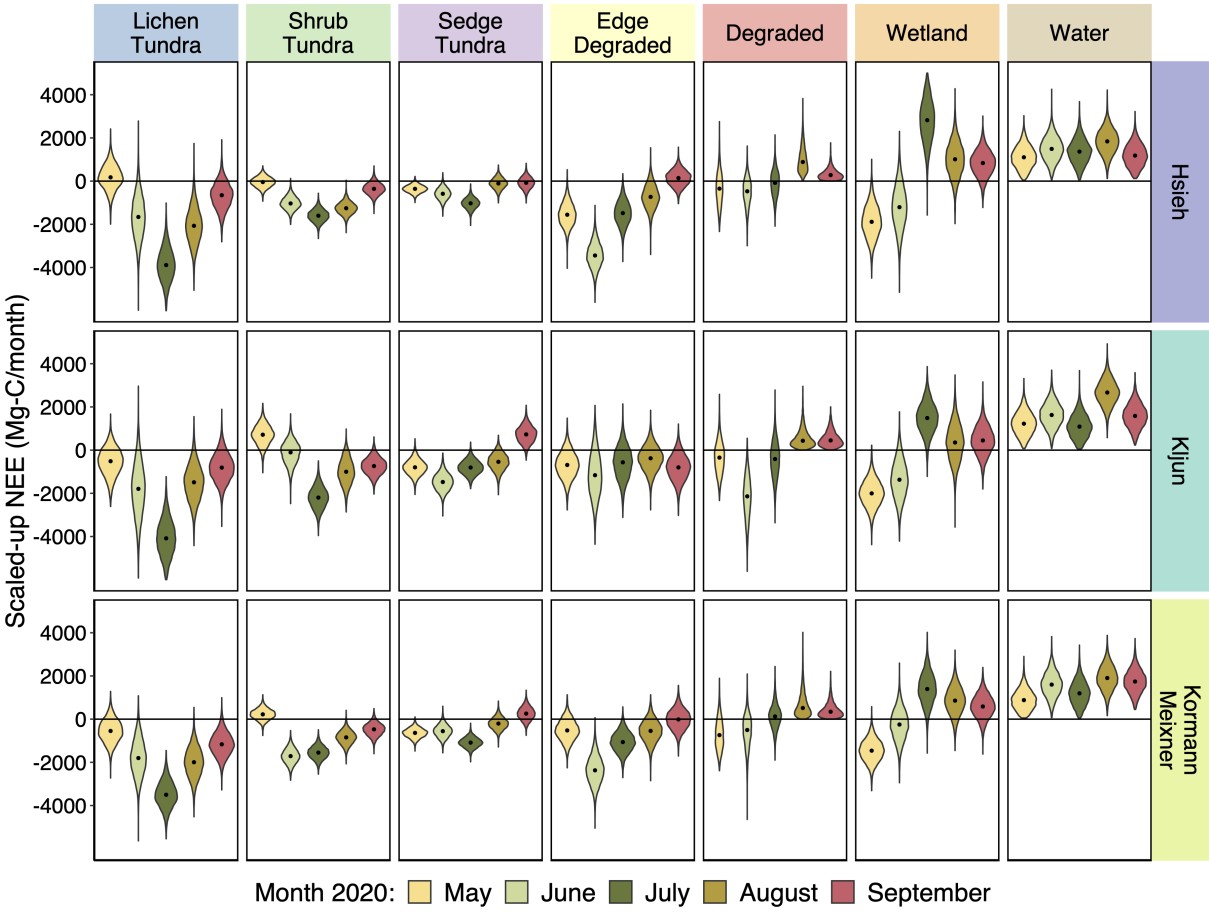

**Figure 6.** Monthly violin plots of 2020 growing season NEE carbon budgets by landcovers (columns) in the complex map for each footprint model (rows). Distributions for violin plots are derived from posterior distributions of predicted NEE fluxes scaled by their landcover areas in figure 1. Black dots indicate medians.

flux tower. However, we were able to capture both plant-mediated carbon fluxes and ebullition in addition to diffusive fluxes, as well as describe the broad seasonal trends in carbon emissions using this approach.

### 3.5 Regional carbon scaling

Landcover carbon budgets were summed to regional carbon budgets and compared to carbon budgets estimated using the Bayesian and MDS homogeneous approaches (Fig 7). For May, August, and September, there were no detectable differences in total NEE between the three footprint models, with only small differences in June and July. Regional $CH_4$ budgets were similar, with most months overlapping between the three footprints, and small differences in July and August. There was also very little difference in median total NEE budgets between the simple and complex map solutions. For most months

the complex map solutions were slightly more uncertain, a consequence of estimating almost twice as many parameters. The





exception was May, where the simple map was more uncertain likely because grouping all tundra vegetation as one class was a poor assumption for that time. Despite the differences in methodologies, the two homogeneous approaches (Bayesian models and MDS) resulted in very similar NEE budgets to one another when scaling up to the region (Fig 7).

In contrast, all homogeneous scaled-up carbon results were quite different from any heterogeneous result, regardless of
footprint model or landcover map choice. While the homogeneous approaches were worse at predicting to withheld gaps in the tower observations (Fig 3), the differences in RMSE were small. However, the consequences for scaling were large. At every part of the growing season the homogeneous NEE overestimated the carbon sink relative to the heterogeneous NEE. This overestimation was smaller towards the shoulder seasons in May and September and larger in June, July, and August. Assuming homogeneity at this site meant approximating the same diurnal cycle of NEE everywhere that averages over different sink and
source strengths in the landscape. Similarly, assuming a homogeneous landscape when scaling up an average $CH_4$ flux led to consistently underestimating the regional $CH_4$ emissions (Fig 7). Because landcovers that were hotspots of emissions were farther from the footprint influence peaks, while those that exhibited larger carbon uptake were more often nearer the peaks, applying the tower flux to the region without accounting for footprints resulted in too much carbon uptake in NEE and too little carbon emissions in $CH_4$.

Throughout the growing season, incorrectly assuming a homogeneous landscape, regardless of gap-filling methodology, resulted in nearly doubling the NEE growing season carbon sink while nearly halving the $CH_4$ emissions (Fig 7f). If we had assumed a homogeneous landscape we would have determined the region to be a net growing season carbon sink even after accounting for $CH_4$ emissions as $CO_2$-equivalents (89% CI: -9960 to -11919 Mg-C, Fig 7f). By deriving the heterogenous landscape fluxes, we instead find that growing season $CH_4$ emissions more than offset the $CO_2$ growing season sink (89%
CI: 1774 to 6403 Mg-C, Fig 7f), an outcome that has been described in other tundra ecosystems as well (Kuhn et al., 2018). Other tower flux sites might see similar or opposite results from accounting for footprint heterogeneity (Griebel et al., 2016; Giannico et al., 2018; Reuss-Schmidt et al., 2019). For example, if a site had landcovers with low carbon uptake or high carbon emissions located near the peak of footprint influences, while more abundant areas of high carbon sinks or low carbon emissions were at the edges of footprint influence, then the resulting carbon budget from scaling while assuming homogeneity
would overestimate carbon emissions. Unaccounted-for heterogeneity such as this could help explain the mismatch between bottom-up and top-down carbon budgets (Thornton et al., 2016; Saunois et al., 2020).

### 3.6 Uncertainty

Uncertainties in model fit, both for respiration, GPP and constant fluxes, were carried through into uncertainties in NEE and $CH_4$ gap-filling and scaled up carbon budgets. Sources of this uncertainty include times and locations where the deterministic
models used here were over-simplifications and failed to capture other important processes affecting carbon cycling. Instances where the landcover maps were not accurate delineations of carbon cycling are also included in this uncertainty. For example, if an underlying gradient of soil moisture were causing different $CH_4$ fluxes within a vegetation type this would lead to greater uncertainty after the aggregation of footprint influences over the categorical map used here. Not all landcover carbon budgets were equal in terms of uncertainty; for example, among the tundra vegetation types, NEE fluxes from edge of degraded areas





**Figure 7.** Monthly total (a-e) and 2020 growing season total (f) carbon budgets for each gap-filling technique. Boxes are the median and interquartile range (IQR), whiskers are 1.5*IQR, for the Bayesian model gap-filling carbon budgets using the three footprint models over the complex and simple landcover maps, and without considering footprints and assuming a homogeneous landscape. The solid red line indicates MDS (marginal distribution sampling) homogeneous gap-filling NEE budgets. The dashed red line indicates the diurnal average CH4 gap-filling homogeneous budgets. Note the scales on y-axes are different between panels to highlight the comparability of footprint models and landcover maps within months.





were the most uncertain, followed by shrub, lichen, and sedge (Fig 4, e.g. Kljun-June standard deviations: 0.061, 0.031, 0.030, and 0.016 kg-C month$^{-1}$ m$^{-2}$ respectively). After scaling up to the region in (Fig 1), lichen NEE carbon budgets were the most uncertain due to their larger area in the region, followed by edge of degraded permafrost, shrub, and then sedge tundra (Fig 6, e.g. Kljun-June standard deviations: 1199, 1009, 516, and 388 Mg-C month$^{-1}$ respectively).

    Degraded permafrost NEE and $CH_4$ fluxes had the most uncertainty (Fig 4, Fig S9). This was likely due to a combination

of their small extent and influence in the footprint supplying less signal to the tower fluxes, as well as the deterministic models over-simplifying carbon processes. In this case, the fluxes from degraded permafrost were distinctive enough they could be determined while un-mixing the tower fluxes despite their small area in the footprints. Within the areas of degrading permafrost there was heterogeneity in vegetation and surface water on a scale smaller than the resolution of the landcover maps, as well as more temporal dynamics related to hydrology. Similarly, NEE and $CH_4$ fluxes from water had relatively large uncertainty due

to estimating an average flux rather than presenting diffusive, plant-mediated, and ebullitive fluxes deterministically. In lieu of representing these processes explicitly, our simpler models had greater uncertainty. The uncertainty around degraded NEE and $CH_4$ fluxes had a smaller impact on NEE carbon budgets than other landcovers (Fig 5, Fig 6) due to the small area of degraded locations in the landscape (Fig 1). Since all of the uncertainties in gap-filling fluxes and partitioning NEE were carried through into carbon budget estimates, we lose nothing from including these small areas of heterogeneity despite not representing them

as well as other landcovers.

### 3.7 Applications and limitations of un-mixing eddy covariance fluxes

Future applications of the flux un-mixing approach demonstrated in this study could incorporate spatially explicit drivers such as soil moisture and soil temperature, as well as more specific prior information from chamber fluxes. Doing so would further reduce uncertainty in landscape carbon fluxes. Seasonality could be represented through spatially explicit and temporally vari-

able drivers such as LAI or solar induced fluorescence (SIF). Interannual variability could be investigated using a hierarchical model structure by, for example, fitting an underlying distribution of a vegetation-type specific $Q_{10}$ from which each year's specific $Q_{10}$ is drawn. This study method interpreting eddy covariance fluxes could also be useful in sites with nested towers, multiple instrument heights, or where instrument heights have changed over time. Flux data from such circumstances could be analyzed concurrently, since each observation is a function of an explicit footprint distribution. Thus, it would not matter if

instrument height or position were different between observations.

    This method of un-mixing eddy covariance fluxes relies upon accurate footprint influence maps with sufficient variability over the heterogeneity in the landscape. The analysis in this study assumes the footprints were observed perfectly, i.e., footprint influence is not a random variable. For this reason, we recommend always using an ensemble of footprint models. However, for sites where the assumptions of footprint models are not met, and footprint influence maps are likely to be more error prone, this

study's methodology will not work. Examples of such cases might include sites with instrument heights close to canopy heights where the effects of the surface roughness sub-layer are a concern, or anywhere the assumption of horizontal homogeneity in turbulence is invalid. In addition to valid footprint influences, this method requires variability in footprints. When footprint influences are aggregated over a landcover type for un-mixing, there needs to be enough differences between observations to





avoid an underdetermined dataset, where finding a solution to landcover specific fluxes won't be possible. Sites with consistent
wind directions and atmospheric stability that result in very similar footprints between observations could have this issue.
Small changes in the peak influence location could create enough variability between observations, even with consistent wind
directions, depending on the position and scale of heterogenous landcover patches at the site.

## 4   Conclusions and implications

We compared multiple footprint models and landcover maps in our analysis to investigate their effects on un-mixing landcover
carbon fluxes. While the Kljun footprint model was the most consistent in determining fluxes when comparing outcomes using
simple and complex landcover maps, there was no clear, best footprint model. We recommend including all three footprint
models as an ensemble when interpreting eddy covariance fluxes. Flux estimates based on the more complex landcover map
captured important differences in seasonality in tundra vegetation carbon fluxes. However, there were only minor differences
in regional growing season carbon budgets between the two landcover maps, and using the more complex map had trade-offs
such as greater computation time and uncertainty due to increasing the number of parameters. Investigating carbon fluxes using
multiple landcover maps allows for informed lumping of landcover classes based on the resulting fluxes and the investigator's
research questions.

Eddy covariance towers provide a wealth of high frequency flux data with large spatial extents. However, tower fluxes
are under-utilized or potentially misleading if footprints are not taken into account in heterogeneous landscapes. We have
demonstrated an approach to un-mixing tower NEE and $CH_4$ fluxes from heterogeneous tundra, which provided detailed
interpretations of landscape carbon cycling such as the detection and quantification of hot spots of carbon emissions and
different timing in peak carbon uptake and senescence in tundra vegetation. We find that methods that consider footprint
influences during gap-filling NEE fluxes were more accurate at predicting missing NEE fluxes than methods that assume
landscape homogeneity. By using a Bayesian approach, we were able to quantify and compare uncertainties between carbon
fluxes from different landcover classes. These uncertainties were carried through when gap-filling and scaling-up, providing an
intrinsic estimate of uncertainty for the resulting carbon budgets. The consequence of assuming homogeneity in the landscape
when gap-filling and scaling-up instead of using landcover-specific carbon fluxes was substantial: over the growing season
(May-September) the homogeneous growing season budgets had half as much $CH_4$ emissions and twice as much net $CO_2$
uptake. Accounting for landscape heterogeneity in carbon fluxes from eddy covariance towers could reduce uncertainty in
bottom-up carbon budgets and the mismatch with top-down carbon budgets.

*Code and data availability.* Eddy covariance flux data, summarized footprint influences, and analysis code are located in the repository:
https://github.com/arctic-carbon/heterogeneous_C_fluxes. Code for the three footprint influence models and reprojecting and summarizing
over the landcover maps are located in the repository: https://github.com/arctic-carbon/eddy-footprint.





*Author contributions.* SMN proposed the study site and instrumentation. SML and SMN set up the tower and instrumentation and maintained
and collected data. SML processed data, developed code, and performed the statistical analyses. SML prepared the manuscript and with
contributions from all co-authors.

*Competing interests.* The authors declare that they have no conflict of interest.

*Acknowledgements.* The authors acknowledge and are grateful for the opportunity to conduct research on the traditional land of the Yup'ik,
who have stewarded this land through many generations. Our work would not have been possible without the support of the Yukon Delta
National Wildlife Refuge, U.S. Fish and Wildlife Service. This study was supported with funding from a National Aeronautics and Space
Administration FINESST grant (80NSSC19K1301) to SML, the Gordon and Betty Moore Foundation (grant #8414) and Woodwell Climate
Research Center Fund for Climate Solutions to SMN, and National Science Foundation Office of Polar Programs grant (#1848620) to RC.
This study was part of the NASA Arctic-Boreal Vulnerability Experiment.



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
