# Peer review of "Resolving heterogeneous fluxes from tundra halves the growing season carbon budget"

_Biogeosciences, 2023_

## Author Comment (AC1)

Please find our response to the reviewers below. We have copied the reviewer comments and our responses are in blue, with the revised text and line numbers further indicated in italics below our responses.

We appreciate the thoroughness of the reviewers' comments and believe they will improve the manuscript. We have taken significant steps to address the reviewers' comments. In addition to addressing all of the minor comments, we have added several figures that will help communicate our results. We have included in figure 1 the mapped region surrounding the EC tower. We have added a figure (S2) demonstrating no relationship between methane fluxes and environmental variables measured at the EC tower. We have added a figure (S4) to help visualize the predicted and observed EC fluxes both as a timeseries and as 1:1 plots. Major changes to the text include an expanded discussion in section 3.7 and statistical differences or similarities indicated in a consistently quantified way via Bayesian 89% credible intervals.

We believe these changes in response to the reviewer's comments have significantly improved the manuscript, and we thank them for their effort.

Reviewer Comment #1:

This manuscript addresses an important problem that is often ignored when analysing data collected with the eddy covariance (EC) technique, a key method for measuring fluxes between ecosystems and the atmosphere. This problem relates to the representativeness of measurement data, which can be compromised if the measurement site is heterogeneous with respect to vegetation and other land cover properties affecting the fluxes. Heterogeneous small-scale flux distribution makes it difficult to derive an unbiased budget estimate for the study area because the EC measurement unevenly weights different land cover elements. It is also difficult to differentiate between land cover types as the measurement represents a spatially integrated flux, making interpretation and regional upscaling of the data challenging.

Ludwig et al. tackle these challenges by developing a statistical technique that can, in their words, 'un-mix' the EC flux measurement. They use this Bayesian method to estimate the contribution of different land cover types to the carbon dioxide and methane fluxes measured at their tundra site in Alaska. As part of the analysis, they compare three different flux footprint models and two data gap-filling approaches. Furthermore, in the context of spatial upscaling of fluxes, they assess the effect of land cover map complexity.

The study is interesting, well focused and suitable for the scope of Biogeosciences. The data appear reliable, and the methods are basically sound. Overall, the analysis is convincing and demonstrate the importance of acknowledging the surface heterogeneity when interpreting EC flux data. However, I find the presentation unsatisfactory in various places and thus recommend major revisions before publication. My concerns are detailed below.

Major comments

(1) The authors say that "numerous footprint models have been developed" (line 35) and "recommend always using an ensemble of footprint models" (line 458). Their analysis is based

on three models, which produce partly diverging results. I have the following questions about this ensemble approach. First, how many models should be optimally used? What were the criteria for choosing these specific models? Did you consider any alternatives? Second, the authors present the results separately for the three models but do not discuss how the results could be combined. How should three different estimates of carbon balance be interpreted, or would it be possible to combine them into a single value and incorporate the variation into an uncertainty estimate?

We selected these three models because they are the most commonly used and referenced for carbon flux interpretations, and all three of these models have been used in other comparisons (Rey-Sanchez et al. 2022). The point about combining the results rather than just comparing between models is very useful. With the Bayesian approach used in this study, we can combine the posterior distributions from each footprint model results and present an overall distribution of scaled carbon that has uncertainty from all three carried through. We have added text to clarify that approach to estimating a scaled carbon budget:

*Line 412-415: We can combine the posterior distributions of scaled carbon from all three footprint model results to calculate a single carbon budget estimate that accounts for across-model uncertainties. Doing this we find that growing season CH4 emissions (mean: 16633, 89\% CI: 15208 to 18212 Mg-C CO2eq) more than offset the CO2 growing season sink (mean: -12512, 89\% CI: -15718 to -9189 Mg-C).*

(2) While Introduction is rich in citations, the same is not true for the Results & Discussion section. Here, the authors should better acknowledge previously published findings. This is especially apparent in Section 3.7 that deals with application of the results. Overall, this is the weakest section of the manuscript and would benefit from a thorough revision; this should be based on tangible results that are related to existing literature (see specific comments).

We have expanded section 3.7 to include examples in the literature that use spatial drivers that could be applied with this footprint un-mixing method, as well as an example of discontinuities in measurement height benefitting from explicit footprint calculations. Detailed responses can be found in the specific comments below.

(3) There are a few places where the text is not supported by data or references.

- lines 163-165: An assumption (about invariability of methane fluxes) central to the statistical approach is presented in the Methods subsection titled "Gap-filling methods". No justification or supporting data analysis is provided for stating the constancy. It would of course be possible to formulate such a hypothesis and try and confirm it later with data (both in an appropriate manuscript section), or simply present it as a model assumption and then discuss how the results depend on that assumption.

Thank you for pointing this out, we agree. We have added a figure to the SI (Fig S2) demonstrating the lack of relationship between observed methane fluxes and air temperature and PAR, as well as citing several examples of other studies that similarly approach methane as a

spatially variable but not temporally variable flux (Rey-Sanchez 2022 et al., Tuovinen et al. 2019, Hannun et al. 2020)

[Figure]

*Figure S2. Methane fluxes from the 2020 growing season (May - September) as a function of PAR (a) and air temperature (b).*

- lines 203-205: How do you know such details about methane flux variation?

Chamber-based fluxes of methane from nearby peat plateau landcovers (tundra vegetation types) were measured in 2017 and these data were archived on the Arctic Data Center. We have added the citation for this dataset.

- lines 217-218: Again, the authors have an idea how methane fluxes behave but do not reveal the source of this information. The Bayesian prior selection should be justified more carefully. Disallowing methane uptake clearly affects the posterior methane flux distribution of the degraded permafrost (Fig. 5).

The more specific prior information used here came from the archived dataset of chamber-based methane fluxes from these landcovers, which is cited here.

- lines 244-245: Here, the flux independency of biometeorological drivers is presented as if it were shown above in the manuscript ("Since the CH4 fluxes did not have relationships…").

We have added a figure to the SI (Fig S2 in the response above) demonstrating the lack of relationship between methane fluxes and biometeorological variables.

*Lines 241-242: Since the CH$_4$ fluxes did not have relationships to any biometeorological drivers such as air temperature or PAR (Fig S2)…*

- lines 290-292: The authors state that "[a]ll Bayesian gap-filling models were able to accurately reconstruct the tower NEE across the growing season as a function of PAR, air temperature, and source attributions" but show no data, which is a major shortcoming. Only the model performance with respect to filling artificial data gaps is presented (Fig. 3). Also, the "Tower

Total" in Figs. S4-8 are scaled fluxes, not actual EC data. Please include a comparison (plots, statistics) of measured vs modelled NEE and CH4 fluxes.

We have added a figure to the SI (Fig S4) showing observed and predicted tower fluxes for the various models to help visualize model accuracy. Please note that the RMSE of the artificial gaps (Fig 3) best demonstrates model accuracy. Best practices for examining model prediction accuracy is to withhold a portion of observations as a testing dataset (the artificial gaps in this study) and use the remaining observations only for training (as we have done) and then predict and evaluate prediction accuracy using only the testing data (RMSE in figure 3). For example, see Vekuri et al. 2023 in *Scientific Reports* for a similar approach to EC model testing and visualization.

[Figure]

*Figure S4. Top: example 10-day timeseries of eddy covariance tower 30-minute fluxes of $CO_2$ observations and predictions from the homogeneous map and complex landcover map using Hsieh, Kljun, and Kormann and Meixner footprint models. Bottom: the entire month (July 2020) dataset of predicted vs. observed tower fluxes for the same set of models.*

(4) Why is the temperature scaling needed in the light response of GPP (Eq. 4)? As far as I can see, none of the papers cited here (line 175) include the temperature (or VPD) attenuation incorporated into Eq. 4. It is also unclear how the temperature scale (Tscale_k) is defined.

Thank you for pointing that out, we have added a line defining Tscale (line 185). We chose to account for a temperature effect this way as it is similar to the parameterization used in PolarVPRM scaling models (e.g, Luus and Lin 2015, Luus et al 2017, Schiferl et al. 2022). We have clarified this in the methods and added these citations.

*T scale_k = (T air_k − Tmin)(T air_k − Tmax)/{(T air_k − Tmin)(T air_k − Tmax) − (T air_k − Topt)$^2$ } (Eq 5)*

*Lines 188-189: GPP is attenuated by temperature using T scale_k, where Tmin = −1.5 ◦C, Tmax = 40 ◦C, and Topt = 15 ◦C (Luus and Lin, 2015; Luus et al., 2017; Schiferl et al., 2022).*

(5) What is the rationale behind calculating the carbon budget as CO2-equivalents where methane fluxes are multiplied by 28 (lines 265-266), which refers to the global warming potential due to a pulse emission over a time horizon of 100 years. What does this quantity indicate in this context? Why a 100-year period? Why a pulse emission approach for a natural ecosystem with fluxes sustained for thousands of years? Also, it is somewhat misleading to call the resulting sum "carbon budget" (that would logically be CO2-C + CH4-C).

We have chosen to do this because these systems are highly subject to climate warming and there is wide interest in interpreting methane emissions on the same scale of impact as carbon dioxide (as seen in Bastviken et al., 2011; Stocker, 2013; Euskirchen et al., 2014; Beaulieu et al., 2020; Skytt et al., 2020). While the 28 global warming potential is not a perfect representation of methane emission impacts and there are other values (ranging from 25- 100) people have used, we chose this because it is a conservative value and common way of interpreting methane emissions in the context of carbon dioxide emissions. Given the disparity in the mass of carbon in CO2 and CH4 emissions, simply adding them together can be misleading and often just reflects the pattern in CO2. We have added citations of other papers that similarly present CH4 and CO2 budget comparisons using CO2-equivalents.

*Line 261-263: CH$_4$ was presented alongside NEE in carbon budgets as CO$_2$-equivalents (CO$_2$-eq) by multiplying CH$_4$ by a factor of 28, a conservative choice among commonly used approximations of relative global warming potentials (Bastviken et al., 2011; Stocker, 2013; Euskirchen et al., 2014; Beaulieu et al., 2020; Skytt et al., 2020).*

Specific comments

lines 4-6: Does 'carbon balance' refer to the Arctic or to 'some locations'?

We have clarified this sentence.

*Line 4-6: With warming temperatures, Arctic ecosystems are changing from a net sink to a net source of carbon to the atmosphere in some locations, but the Arctic's carbon balance remains highly uncertain*

line 64: This in inexact. The synthesis excluded EC measurements only if they could not be attributed to a specific land cover type.

*Thank you for pointing this out, we have rephrased to be more accurate.*

*Line 63-64: A recent synthesis of circumpolar CH4 fluxes excluded EC measurements that could not be attributed wholly to wetland or waterbody sources*

line 64: I have not previously seen the term 'unmix' (or 'un-mix') being used to mean estimation of land cover specific fluxes. Please define the meaning.

*We have added a line defining "un-mix" in the context of eddy covariance fluxes.*

*Line 75: We used footprint models and landcover maps to unmix EC fluxes into constituent landcover fluxes in heterogeneous tundra in the Yukon-Kuskokwim (YK) Delta, Alaska*

lines 66-67: You should refer to regional budgets rather than individual ecosystems, which do not require a network.

*Multiple flux observation sites are required to characterize the variability within ecosystems. We have removed the word 'network' in case that implies a regular grid of EC towers.*

*Line 65: Arctic ecosystems in particular require representative carbon flux observations to accurately derive seasonal and annual budgets.*

lines 67-69: Rantanen et al. only studied warming, not carbon dynamics or related climate feedbacks.

*We thank the reviewer for catching that. We have included additional references in this line related to carbon dynamics and feedbacks (Schuur et al. 2015)*

line 83: I do not find any discussion on "arctic carbon feedbacks".

*We discuss Arctic carbon feedbacks in lines 503-509: The consequence of assuming homogeneity in the landscape when gap-filling and scaling-up instead of using landcover-specific carbon fluxes was substantial: over the growing season (May --- September) the homogeneous carbon budgets had half as much CH4 emissions and twice as much net CO2 uptake, greatly overestimating the carbon sink in the region and potential negative feedback to climate from carbon emissions. Accounting for landscape heterogeneity in carbon fluxes from EC towers could reduce uncertainty in bottom-up carbon budgets and the mismatch with top-down carbon budgets.*

lines 98-100: How were the land cover classes defined? Did you have any vegetation or soil survey data available? Did you validate the land cover maps?

The details of how the landcover map was created, landcover classes defined and validated, are in the reference cited where the map was first published (Ludwig et al. 2022) and in the archived landcover map on the ORNL DAAC which was cited as well (Ludwig et al. 2023).

See lines 94-95: *We used a landcover map developed by Ludwig et al. 2022 to characterize the EC tower location and a nearby region of unburned tundra used for scaling up carbon budget (Ludwig et al. 2023a).*

Table 1: It would be useful to show similar percentages for the average footprint-weighted land cover proportions during the study period, estimated with different footprint models.

The full distributions for the footprint-weighted landcover proportions for each of the three footprint models are in Figure 2 and in the SI. We chose to display the tower area proportions instead of footprint-weighted averages since none of the distributions are Gaussian and displaying the average is not a useful metric or central-tendency in this case. We believe that including the actual distributions where the range and variance are clear is a better way to communicate this.

line 127: What criterion was used for nonstationarity?

We used the Foken et al. 2004 method, as is cited in line 126 (implemented in EddyPro). This method assigns integers to indicate fluxes that pass or fail these QA/QC checks on the high frequency data. This method is a widely used in established EC networks (e.g. Ameriflux and Fluxnet).

line 127: What is 'rssi'?

We have removed the acronym for clarification.

line 128: u* should be defined. You could include the CO2 flux vs u* analysis in the supplement.

We have explicitly defined u*. All of the QA/QC and filtering steps, including the figures demonstrating where the u* threshold is located, are in publicly available code in a repository linked in our code and data availability statement.

*Lines 127-128: ...low turbulence (friction velocity (u*) < 0.1 ms−1 threshold was chosen where CO2 fluxes were independent of u*).*

lines 128-129: I'm not sure if I'd call an energy balance closure of 70% 'good'. It is a typical value, probably rather low than high. Data or citation needed if this is included.

We have replaced 'good' with 'typical'.

Figure 1: Please indicate the location of the EC tower.

The tower is located in a section of unburned tundra just SW of this lat-long bounding box. We have added the mapped 300m circle surrounding the tower (as described in Table 1) as another panel. We chose not to extend the bounding box in panels a-c further SW because there are various fire scars nearby are not being mapped or scaled-to, and would be an unnecessary confusion to include visually in Figure 1.

Section 2.4: Why is this section written from the gap-filling point of view only? The methods presented here are central to flux 'unmixing', which is the key topic of the manuscript.

We have included 'unmixing' in the section title to be more inclusive, as well as clarified below that the gap-filling models can be understood more broadly as models that predict carbon fluxes.

*Lines 155-156: We compared several modeling approaches for predicting and gap-filling the EC NEE timeseries.*

lines 171-172: The solution of model equations is presented in Section 2.4.2 and is irrelevant here.

We have removed this line from the methods.

lines 190-198: This sounds like discussion. Especially the last sentences would improve the actual discussion section.

We thank the reviewer for this point. We have moved this paragraph to section 3.7 where it does indeed improve the discussion section.

lines 207-208: It sounds a bit strange to present ordinary least squares as the only alternative to MCMC simulations. There exists a plethora of fitting methods suitable for non-linear relationships.

We have rephrased this sentence and included a reference to Rößer et al. 2019, as an example of a non-linear solution.

*Line 204-207: We chose this method partly because unmixing approaches such as ordinary least squares (as used by Tuovinen et al. (2019)) are not applicable with the non-linear relationships used here between $CO_2$ and air temperature and PAR, and non-linear ordinary least squares (as used by Rößger et al. (2019)) assumes normal distributions for parameters and error variance, which is often not the case.*

lines 214-216: I do not understand this explanation for using uninformative priors. Please clarify.

We have added clarification. We chose to show that the method works with reasonable certainty without needing to provide specific prior information. More specific priors would reduce the uncertainty in posterior distributions, which is useful for reducing uncertainty in scaled carbon budget, but is not strictly necessary here.

*Lines 211-213: We used the latter approach to NEE priors for this study to demonstrate the impacts of unmixing EC tower NEE on gap-filling accuracy and bottom-up scaling while using the fewest assumptions.*

lines 245-246: Unclear what is meant by "calculating the average CH4 flux for each half hour period". Aren't you rather calculating the daily (and monthly) averages?

We have rephrased to clarify. We are calculating the monthly average CH4 flux of each half-hour of a diurnal cycle. For example, the average of all July CH4 fluxes at 16:30 was used to scale up CH4 fluxes for 16:30 all days in July.

*Lines 242-243: we estimated monthly budgets by calculating the monthly average CH4 flux for each half hour of the diurnal cycle, and then applying these averages to each day within the month.*

lines 252: What does the percentage range represent?

There was a little variation between months in the total amount of withheld data using random stratified gaps. We have clarified the percentage range is between months.

*Line 249: Between 15 — 20% of the timeseries of each month…*

lines 253-254: Was there any rationale behind this gap scenario; i.e., does it reflect the gaps found in real measurement data?

Yes, these gap lengths were chosen to have similarities to commonly found real gap sizes. A detailed analysis of gaps in EC timeseries (such as in Kim et al. 2019) is beyond the purview of this study.

line 277: Why do you find the agreement between the footprint models 'surprising'?

We have edited this sentence to be less vague.

*Lines 276-277: There was a fair amount of agreement between the three footprint models, with the majority of footprint influences close to the 1:1 line on regressions between model types (Fig 2, Fig S3).*

line 283: Why 'likely'? You can check these "distinct periods" from the data.

We have rephrased line 283 to be less ambiguous.

*Line 282: There are also distinct periods of larger differences between footprint models, for example when the peak footprint influence was near the boundary between two landcover types*

lines 305-306: But, as you say, MDS is biased in high latitudes.

This was worth mentioning because MDS is still by far the most widely used gap-filling method including in high latitudes, and its convenience could outweigh any cons for some investigators.

lines 307-308: In many cases, the RMSE differences between models are larger than the differences between land cover maps, so it is not clear what 'overlapped' implies. The comparisons should be based on a statistical criterion.

We have added letter designations to Fig 3 to show which RMSEs have credible intervals that overlap. However, in Bayesian approaches it is unusual to use an arbitrary threshold, such as p-values, to decide if something is significant or not. In the frequentist realm for example, if two 95% confidence intervals overlap they are considered not significantly different and the amount that they overlap (or don't overlap) has no meaningful interpretation. On the other hand, for Bayesian credible intervals one can interpret the amount of overlap or separation as a measure of similarity or difference. For example, one could compare credible intervals that overlap by 50%, 5%, or have separation of 10% or 150%, and these differences are meaningful in a way that has no analog in frequentist confidence intervals. Credible intervals that have some overlap would not necessarily be interpreted to have statistically insignificant differences the same as one would interpret overlapping confidence intervals.

lines 308-309: Comparing the RMSE medians, this is true only for the monthly CH4 data, but for the growing season CH4 total the Kormann-Meixner model performs better than the Kljun model. For NEE, the K-M model has the highest RMSE for two out of five months for both land cover maps, so "more often than not" does not check out. Considering this and observing the large dispersion, I would not imply that the K-M model showed the worst performance.

While the KM model does perform worse more often than the others, the margins on these differences are small and we recommend using all three models to capture footprint model uncertainty in interpreting EC data. Since this study is not attempting to validate footprint models against known fluxes (such as Rey-Sanchez et al. 2022), but rather capture the impacts of footprint model choice on scaling carbon, we have rephrased this discussion point.

*Lines 307-309: None of the three footprint models consistently performed better in terms of RMSE, and for most outcomes, the Bayesian 89% CI for their RMSEs overlapped (Fig 3, Fig S5). Given that none of the three footprint model types quantify their uncertainty, we continued to evaluate all three as an ensemble of footprint models that represents the range in footprint influence outcomes.*

lines 318-319. While there clearly is the stated difference for degraded permafrost in June, without supporting statistics it is not possible to generalize the model performance for other landcover types and for other months. I also do not understand why the footprint models are not assessed in relation to the measured 30-min EC fluxes.

We rephrased this section to clarify. The Kljun model is the best at estimating a consistent landcover flux for degrading permafrost between the simple and complex landcover maps. Each landcover class is estimated simultaneously since the weights must sum to 100%, so a

discrepancy in degrading permafrost means it is being offset in another landcover, and the performance across landcover classes need to be considered as a set.

*Lines 317-319: The Hsieh and Kormann and Meixner models were notably inconsistent for degraded permafrost for June and July (Fig S7-8), while the Kljun footprint model was always distinctly consistent (Fig S6-10)*

The footprint models were assessed in relation to the measured 30-min EC data. Data were withheld via artificial gaps and used to test the models' abilities to predict the 30 min EC fluxes (Fig 3).

lines 326-339: Here, the presentation would benefit from more quantitative comparisons (instead of inexact expressions such as "aligns with", "similar to" and "similar in range").

We have included the maximum value of CO2 uptake amongst tundra vegetations in the text.

*Lines 325-326: had higher peak carbon uptake (-0.342, -0.266, -0.308 kg-C month−1 m−2, for Hsieh, Kljun, and Kormann and Meixner) than sedge and lichen tundra (-0.175, -0.175, -0.139 kg-C month−1 m−2, for Hsieh, Kljun, and Kormann and Meixner) (Fig 4).*

We used 'aligns with previous studies that have found...' appropriately.

lines 342-343: Posterior distributions (Fig. 5), parameter estimates (Supplement) and actual flux data (Fig. S1b) show that you cannot rule out the possibility of occasional CH4 uptake by tundra. Comments on this would be useful.

The few sporadic negative fluxes of methane in Fig S1b are noise and not true uptake. Eddy covariance timeseries are noisy, particularly methane fluxes, and removing negative values is not recommended because this will lead to a positive bias. Neither are the parameter distributions indicating uptake, just tails of distributions that cross zero, or the parameters are functionally deriving a zero or very small flux whose distribution includes zero. The chamber data (cited above) does not show any evidence of methane uptake for the range of ecosystems near the tower.

lines 358-362: This text deals with NEE (Fig. 4), not carbon. Please rephrase.

We have replaced 'carbon' with 'carbon dioxide'

line 363: "less prevalent", please quantify; "wetland could not to converge", unclear meaning.

Thank you for pointing this out. We have clarified this discussion point.

*Lines 363-364: The landcover map used in this study identified two types of wetlands, one much more prevalent near the EC tower than the other. Attempting to use both wetland types failed, as the parameters for the less prevalent wetland could not converge.*

lines 371-372: carbon or CO2-C?

This sentence is correct as is with 'carbon'.

lines 387-388: How do you know you have a representative sample of these fluxes? The footprint coverage of water areas is low and often zero. How does this relate to the sporadic nature of ebullition and the seasonality of plant-mediated gas transport? And earlier in the same paragraph you concluded that the water fluxes are likely overestimated.

We agree, the water fluxes here are not likely to be representative. This is why we discuss the approach in Ludwig et al. 2023, and reference it as a better method for water carbon flux scaling. The method described in Ludwig et al. 2023 deals with the representativeness of various waterbodies, watersheds, water chemistry, etc. As we described in this paragraph, it is well documented in many ecosystems and specifically in this region that smaller waterbodies have larger carbon fluxes, as such using the fluxes derived from the small waterbodies near the tower to scale to all waterbodies throughout the region is likely an overestimate. The sporadic nature of ebullitive and plant mediated fluxes and their low contribution in footprint weight is likely why here we could estimate a mean and variance for methane fluxes but nothing more mechanistic, and this is explained in the methods (lines 185-188).

lines 390-391: Fig. 7 shows the CO2-eq. budget, not the carbon budget.

Please see in the methods:

*Line 261-263: CH4 was presented alongside NEE in carbon budgets as CO2-equivalents (CO2-eq) by multiplying by a factor of 28, a conservative choice among commonly used approximations of relative global warming potentials (Bastviken et al., 2011; Stocker, 2013; Euskirchen et al., 2014; Beaulieu et al., 2020; Skytt et al., 2020).*

lines 391-393: How do you define "detectable differences" and "small differences"?

We have updated this discussion using to clarify 'overlapping 89% credible intervals'. Posterior distributions with no overlap in credible intervals are denoted with different letters in Fig 7.

line 395: Why would increasing the number of model parameters increase uncertainty? I would have assumed that this makes the model more flexible and thus decreases uncertainty.

Increasing uncertainty with increasing complexity is a common occurrence across modeling. See Puy et al. 2022 (https://www.science.org/doi/10.1126/sciadv.abn9450) for a nice summary.

line 398: Not in September.

This is likely because MDS is most biased in shoulder seasons, where light and temperature distributions are even more skewed. We have added that comment to the discussion.

*Line 400-403: In months closer to the shoulder season (May and September), the distributions of light and temperature are more skewed, which is a source of bias in the MDS method and could explain the slight differences in the MDS and Bayesian homogeneous results for those months.*

lines 413-415: It is misleading to use Mg-C as the unit for the CO2-eq budget. Note that Kuhn et al. (2018) you cite here presented carbon balances (CO2-C + CH4-C), not CO2-eq balances. Using the CO2-eq. concept implies that the ecosystem has a warming effect on climate.

Using $CO_2$-eq is a common way of presenting side by side methane and carbon dioxide emissions (as seen in Bastviken et al., 2011; Stocker, 2013; Euskirchen et al., 2014; Beaulieu et al., 2020; Skytt et al., 2020).

Kuhn et al 2018 found that including commonly overlooked small pond $CO_2$ and $CH_4$ emissions offset a large portion of the wetland carbon sink. We find a similar result, though our study looks at multiple types of small-scale heterogeneity not just the presence of small ponds. We have rephrased so as to be clear we are not implying that Kuhn et al 2018 also reports methane in CO2 equivalents.

*Line 420: Similarly, (Kuhn et al. 2018) found that accounting for emissions from commonly overlooked small ponds offset much of the wetland carbon sink in Northern Sweden.*

lines 417-420: This text basically repeats what is written in the previous paragraph.

The previous paragraph described the results in this study in the context of this sites spatial arrangement of sources and sinks. Whereas in this sentence we explain that a different positioning of landcovers could have resulted in the opposite pattern, which is useful to understand in the context of comparing bottom-up and top-down scaling since the discrepancy there is one sided (bottom-up methane budgets are higher than top down). We have rephrased for clarity.

*Lines 424-425: A heterogeneous site with low carbon uptake or high carbon emissions located near the peak of footprint influences would overestimate carbon emissions when scaling assuming homogeneity*

line 434: In the previous paragraph, you said that the edge of the degraded areas was the most uncertain land cover type.

Line 434 states that NEE fluxes from the edge of degraded areas were the most uncertain *among tundra vegetation types*.

line 440: This obviously depends on the accuracy of the deterministic approach. Do models and data exist for this?

Yes, for example see the discussion of water carbon fluxes earlier citing Ludwig et al. 2023. That study uses $CO_2$ and $CH_4$ measurements from hundreds of waterbodies in the region to scale

carbon emissions as a function of waterbody size, color, shape, and watershed size and landcover.

lines 447-455: This paragraph is basically a list of potential improvements to the method presented in the manuscript rather than a discussion of actual applications of the method as it is. Citations are needed to indicate previous occurrence of these ideas.

We have expanded this section:

*New Lines 453-476: Heterogeneity within EC tower landscapes is a common problem, and employing this flux unmixing approach at sites such as those identified by (Chu et al. 2021) could improve accuracy in scaling carbon budgets and bench-marking models. Several studies have used summed spatial variables after weighting by EC footprints to relate to EC flux observations (Reuss-Schmidt et al. 2019, Xu et al. 2017, Metzger et al. 2013). While a useful way to incorporate heterogeneity, this approach reduces meaningful variation of spatial variables within footprints to single non-unique results. For example, there are multiple combinations of footprints weights and values of the spatial drivers that could result in the same weighted sum. Statistically unmixing fluxes could yield more informative relationships to spatial drivers.*

*Future applications of the flux unmixing approach demonstrated in this study could incorporate spatially explicit drivers such as soil moisture and soil temperature, as well as more specific prior information from chamber fluxes. Doing so would further reduce uncertainty in landscape carbon fluxes. Seasonality could be represented through spatially explicit and temporally variable drivers such as solar induced fluorescence (SIF) ( Schiferl et al. 2022). Interannual variability could be investigated using a hierarchical model structure by, for example, fitting an underlying distribution of a vegetation-type specific Q10 from which each year's specific Q10 is drawn. This method of interpreting EC fluxes could also be useful in sites with nested EC towers, multiple instrument heights, or where instrument heights have changed over time (e.g. (Klosterhalfen et al. 2023). Flux data from such circumstances could be analyzed concurrently, since each observation is a function of an explicit footprint distribution. Thus, it would not matter if instrument height or position were different between observations.*

*An alternative model structure for GPP was investigated that uses leaf area index (LAI) as a driver (Shaver et al. 2007). In lieu of field-based LAI data, we used a timeseries of NDVI from cloud-free Sentinel-2 imagery and the empirical relationship to LAI from pan-Arctic tundra described in (Shaver et al. 2013). The LAI-version GPP model failed posterior predictive checks for most months of data, and was not further pursued. This failure is likely because the approximation from NDVI was a poor representation of LAI for this site, particularly during May, August, and September where sub-pixel water presence could lead to erroneous NDVI and LAI. Furthermore, lichen and moss species dominated the vegetation biomass on peat plateaus and LAI may not be an appropriate metric in such cases. However, a spatially resolved driver such as LAI might be effective in applications for unmixing NEE at other sites.*

lines 456-467: This paragraph is more relevant to the applicability of the method than the previous one. However, the discussion is rather superficial and requires more quantitative results.

Would it be possible to specify the required "variability in footprints" (line 462) and what kind of "differences between observations" are 'enough' (line 463)? In addition, the conclusion that violation of model assumptions increases uncertainty is rather trivial (line 459); isn't heterogeneous turbulence a problem that concerns the EC measurements in the first place (if the measurements do not fulfil assumptions, then any related modelling is obviously redundant) (lines 461-462); suggesting that "consistent wind directions and atmospheric stability" could be a problem requires a real-world example (lines 464-465).

Assessing the variability in footprints and the corresponding minimum differences in landcover fluxes needed to be able to statistically un-mix the fluxes is a site-specific question. Generalizing the required variability for other sites is beyond the scope of this study but we have made our code available in a public repository and welcome others to try this approach at their site. Yes, homogeneous turbulence is an assumption for EC, but it is worth explicitly stating here since there are EC towers in non-ideal landscapes. The particular atmospheric conditions and invariance in wind directions that could make this method difficult to implement are a site-specific problem, and evaluating that for other EC locations is also beyond the scope of this study.

Table S10: A similar table showing monthly mean (SD) fluxes for all land cover types would be useful.

The full distributions for each month and landcover for methane are in Figure S9. Figure 4 shows the full distribution by month and landcover for carbon dioxide, summed by month. Given the strong positive and negative diurnal signal in $CO_2$ NEE, the monthly sum is informative and an appropriate choice instead of the mean.

Technical comments

line 73: Word(s) missing.

 *"difficulties in calculating representative"*

line 79: "compared net ecosystem exchange (NEE) results from CO2 fluxes"; please rephrase.

Clarified to:

*"We compared the net ecosystem exchange (NEE) of $CO_2$".*

While most commonly applied to $CO_2$, the term net ecosystem exchange can apply to any gas (e.g. OCS, CH4, CO, VOCs, etc). Here, in the first use of NEE we specified we are using it in regards to CO2.

line 128: Unit missing.

We have added the unit ($m\ s^{-1}$)

Section 2.3: Incorrect title.

Thank you for catching that, we have corrected the title to 'Eddy covariance footprint modeling'

line 169: Remove "equation 1".

Changed to (Eq 1)

line 202: Remove "equation 5".

Changed to (Eq 6).

line 227: A wrong unit.

We have updated to umol m-2 s-1

Figure 4. Why is the corresponding CH4 plot placed in the supplement? The time unit (month) of the flux is rather uncommon.

A monthly time unit is most appropriate here since each month of $CH_4$ data was trained separately. The time unit of month is also a common way of summarizing carbon fluxes, especially methane (e.g. Miller et al., 2016). We included both the monthly fluxes and landcover area-scaled NEE of $CO_2$ since the magnitudes and uncertainties of each flux are compared in the discussion. For methane, there is little value in including Figure S9 (methane per area) in the main text instead of the supplement, since it has a very similar pattern between landcovers as the scaled version (figure 5).

lines 430-431: The word 'tundra' missing.

At the beginning of the sentence it states 'among tundra vegetation types'

Whole text: There are inaccuracies in writing, some examples:

- Expressions such as "chamber fluxes", "tower fluxes", "tower NEE and CH4" and "eddy tower" sound colloquial.

We have updated the text to consistently use the term eddy covariance (EC) tower fluxes. Specifying 'tower' is useful to distinguish fluxes as modeled or observed at the tower from *landcover* fluxes that are also derived from the EC dataset. "Chamber" fluxes is a widely used and accepted term. For example, see Stoy et al. 2013 paper titled "Upscaling tundra CO2 exchange from chamber to eddy covariance tower".

- Incorrect prepositions: "from May-October", "Between 15-20%", etc.

We have replaced '-' with 'to' and 'and' .

Tables S1-S3: Please explain the distribution notation.

We have added a sentence explaining the JAGS distribution terminology to the SI.

*Tables S1 – S3 include prior distribution information following the format used in JAGS. Those reported here include normal distributions 'dnorm' with parameters mean and standard deviation, and uniform distributions 'dunif' with parameters minimum and maximum range.*

Tables S1-S10: Please unify the number of decimals.

We use a consistent number of significant digits throughout.

Table S3: A value missing.

Thank you for catching that, we have added the missing value.

Reviewer Comment #2:

This paper explores the spatial heterogeneity in fluxes of $CO_2$ and $CH_4$ in a Tundra ecosystem in Alaska. The authors present a novel approach to decompose ("un-mix") the EC signal from the different land covers. The authors find that using gap-filling methods that take into account the decomposed signal from this approach results in better performances than other gap-filling methods, and that scaling up using this decomposition approach can result in up to 2-fold differences in the total fluxes.

I think the use of Markov Chain Monte Carlo (MCMC) simulations to predict the fluxes using footprint decomposition is novel and worth of publication. The manuscript is well-written, well organized and the message is clear. However, some methodological questions need to be addressed first:

Perhaps the most important one is the assumption of constant fluxes of methane (L. 198) from different land covers. This is a big assumption since one would expect seasonal changes in methane driven by temperature and variability among land cover contributions driven by fluctuations in water level. For example, one land cover may have a larger flux than other under low water table conditions, but the relationship may switch under highwater tables. How can this be addressed? Some of these limitations are addressed in the discussion, but perhaps a bit more discussion on the specific limitations for methane flux calculation is needed. More comparison against chamber derived $CH_4$ fluxes from different land covers can enhance this discussion.

We thank the reviewer for highlighting this point. We account for seasonal (monthly) and spatial differences in methane fluxes by deriving different distributions of fluxes trained separately on each month of data for each landcover. We do not have an explicit (sub-monthly) temporally-varying driver of methane, since there was no relationship between methane and the other measured variables at the EC tower. We have added a figure to the SI that demonstrates this (fig S2 included above). Modeling methane as a temporally constant (within a month) but spatially variable flux is commonly done (Rey-Sanchez 2022 et al., Tuovinen et al. 2019, Hannun et al.

2020). We discuss how landcover specific temporal drivers (such as soil moisture and temperature) could improve methane flux un-mixing, but we cannot discuss specifics as those data do not exist for this site concurrent with the flux measurements presented here.

*Lines 461-463: Future applications of the flux unmixing approach demonstrated in this study could incorporate spatially explicit drivers such as soil moisture and soil temperature, as well as more specific prior information from chamber fluxes. Doing so would further reduce uncertainty in landscape carbon fluxes.*

More details about the footprint calculations are needed. It is not mentioned if footprint contours were applied or if all the footprint weights were calculated for the domain of the area shown in Fig 1? This needs more detail. Perhaps a table with summaries of average percent footprint coverage before normalization, fetch at 80% footprint contour, average footprint width, and average distance from the tower to the point of maximum footprint weight, will inform the reader about the footprint differences for each model.

We have added detail about the footprints.

We defined each footprint across a 2000 m x 2000 m area centered on the tower, which was well in excess of 90% influence. This way every footprint even with different extents were well included. The pixels at the tails had effectively no impact since we were using only the summed footprint weights of each landcover, and all footprint models asymptote to zero. Similarly, normalization had little effect since the weights are unitless and all were very nearly 100% contained anyways. There was far more variation in footprints between observations through time than between footprint models, so comparing e.g. average widths between models was not found to be instructive. We aimed to show the range in outcomes for carbon scaling from choosing each model since most studies that use footprint models have picked just one of these three. See Rey-Sanchez et al. 2022 for a good comparison of these three models with a known point source of methane emissions with in the footprints.

*Lines 148-150: Each footprint was modeled 1000 meters in the downwind direction, and 250 meters to either side in the crosswind direction. These values were chosen as they were well in excess of the 90% contours of all footprints (peak influences were < 100 meters and the 90% contours averaged 200 meters from the EC tower)*

The MCMC simulations is a useful approach but how good is it for methane emissions given the assumptions. One key question is how does this method compare to machine-learning gap-filling approaches (e.g. Artificial Neural Networks) that take into account the wind direction but also all the other important environmental drivers?

Wind direction alone is not sufficient to represent heterogeneity within a footprint. At the site used in this study, a NN gap filling approach would not have worked any better since those methods rely on existing relationships between other measured drivers and methane fluxes, which was not observed here. Those methods would likely work better in a homogeneous landscape or in a landscape with heterogeneity that corresponds to clear sectors that can be

captured by wind direction, provided a causal relationship existed between measured environmental drivers and methane fluxes as well.

Specific Comments

What are the references for the equations for Respiration and GPP? (Eqs 3 and 4)

These are cited on line 177 (Williams et al., 2006; Shaver et al., 2007; Loranty et al., 2011).

Fig 1. Is the location of the tower in the center of the map?

We have included a panel with the 300 m radius area around the tower mapped as well, with the tower location indicated (Fig 1). The tower is located on a stretch of unburned landscape surrounded by several fire scars. We chose to scale to this region since it is all unburned and we are not mapping or scaling to the burned areas.

1.  125 Is the time lag removal by covariance maximization necessary for open path instruments?

Yes, although it is generally smaller than those for closed path instruments.

L.127 an RSSI lower than 15% is an extremely low threshold. Can you provide justification for the use of this threshold?

The median value of RSSI for the QA/QC'ed final dataset that was used here was 99.38%. We could have been less conservative about the RSSI threshold and it would have had very little affect on the dataset. This is the threshold used for designating bad values within the LI-7500DS diagnostic output.

L.146 Can you provide more details about the roughness length calculation? It seems rather low. It is also not clear how canopy height was estimated. Was it assumed to be zero?

This roughness length is a typical value for tundra (see McFadden et al. 2003 for a summary of tundra sites that shows similar values for canopy heights and roughness lengths). Monin-Obukhov similarity theory is used to determine roughness length. Under neutral conditions, wind speed is equal to the friction velocity divided by the Von Karman constant, times the log of the measurement height over roughness length. The shrubs are less than a few centimeters tall and sporadic, so a canopy height of 0-10 cm for tundra is common.

*Lines 145-146: We calculated a single roughness length for the site (0.02 m) from the measured wind speed and friction velocity under neutral conditions assuming a logarithmic wind profile and zero displacement height*

---

## Author Response (AR2)

Dear Editor,

Thank you for the chance to respond to the remaining minor comments. We have addressed the reviewer comments and edited the manuscript where changes were requested. Please find our response to the reviewer below. We have copied the reviewer comments and our responses are in blue, with revised test and line numbers further indicated in italics below our responses as applicable.

Thank you,
Sarah Ludwig (on behalf of all authors)

Reviewer comments:
This is a review of the revised manuscript version. The authors have provided response to my original review comments and revised the manuscript accordingly. Overall, I find the response satisfactory and am pleased to recommend publication. I have some further, relatively minor comments the authors may wish to consider (RE-REVIEW COMMENTs below). I do not expect an additional review round.

(1)

REVIEW COMMENT - lines 217-218: Again, the authors have an idea how methane fluxes behave but do not reveal the source of this information. The Bayesian prior selection should be justified more carefully. Disallowing methane uptake clearly affects the posterior methane flux distribution of the degraded permafrost (Fig. 5).

AUTHORS' RESPONSE - The more specific prior information used here came from the archived dataset of chamber-based methane fluxes from these landcovers, which is cited here.

RE-REVIEW COMMENT:

The original comment referred to the following sentence: "We used mostly uninformative prior fluxes for landcovers anticipated to support CH4 emissions by disallowing CH4 uptake for degraded, edge of degraded, wetland, and water landcover classes". The dataset mentioned in the authors' response (Ludwig et al., 2018) does not follow this sentence but the following one, which is about the peat plateau fluxes. The dataset in question contains CH4 flux data only for the peat plateau (divided into lichen and moss patches, 36 data points in total). As far as I can see, there are no flux data for the wetland and water landcover classes. Furthermore, it is not obvious which data included in the dataset correspond to the 'degraded' and 'edge of degraded' classes as the classification is not fully consistent with the manuscript. It would be useful to present a consistent, quantitative summary of the measurement data that were actually used in the Bayesian prior selection process.

Thank you for clarifying. We have added a citation for fluxes from another dataset from water, wetland, and degraded permafrost sites nearby that clearly demonstrate these are methane producing locations. The assumption that wetlands and small ponds do not take up methane is

very reasonable and does not require specific quantification to be summarized here. We have further clarified in the text that these sites are fully saturated soils or open water.

*Lines 215-218: We used mostly uninformative prior fluxes for landcovers known to emit $CH_4$ (e.g. fully saturated soils and open water) by disallowing $CH_4$ uptake for degraded, edge of degraded, wetland, and water landcover classes (Ludwig et al. 2018a).*

(2)

REVIEW COMMENT - lines 244-245: Here, the flux independency of biometeorological drivers is presented as if it were shown above in the manuscript ("Since the CH4 fluxes did not have relationships…").

AUTHORS' RESPONSE - We have added a figure to the SI (Fig S2 in the response above) demonstrating the lack of relationship between methane fluxes and biometeorological variables.

RE-REVIEW COMMENT:

The new figure (Fig. S2) aims to demonstrate that the CH4 flux does not depend on environmental drivers. Perhaps the authors could explain why they have chosen PAR and air temperature as the drivers that would potentially control the CH4 flux (rather than soil temperature and moisture) and why they aggregate the data for the April-September period while the fluxes are modelled on a monthly basis. In any case, the statement that "CH4 fluxes did not have relationships to any biometeorological drivers such as air temperature or PAR (Fig S2)" (line 253) should be revised as the relationship was only tested for these two variables.

Soil temperature and moisture were not measured in wetlands, degraded permafrost, or edge of degraded permafrost during this study period. We have change 'any' to 'observed' in the main text. The figure S2 does not show aggregated data, it shows scatterplots of all observations. Whether within months or across the growing season, there is no relationship between CH4 and PAR or temperature.

*Lines 243: Since the CH4 fluxes did not have relationships to observed biometeorological drivers such as air temperature or PAR (Fig. S2)…*

(3)

REVIEW COMMENT - What is the rationale behind calculating the carbon budget as CO2-equivalents where methane fluxes are multiplied by 28 (lines 265-266), which refers to the global warming potential due to a pulse emission over a time horizon of 100 years. What does this quantity indicate in this context? Why a 100-year period? Why a pulse emission approach for a natural ecosystem with fluxes sustained for thousands of years? Also, it is somewhat misleading to call the resulting sum "carbon budget" (that would logically be CO2-C + CH4-C).

AUTHORS' RESPONSE - We have chosen to do this because these systems are highly subject to climate warming and there is wide interest in interpreting methane emissions on the same

scale of impact as carbon dioxide (as seen in Bastviken et al., 2011; Stocker, 2013; Euskirchen et al., 2014; Beaulieu et al., 2020; Skytt et al., 2020). While the 28 global warming potential is not a perfect representation of methane emission impacts and there are other values (ranging from 25- 100) people have used, we chose this because it is a conservative value and common way of interpreting methane emissions in the context of carbon dioxide emissions. Given the disparity in the mass of carbon in CO2 and CH4 emissions, simply adding them together can be misleading and often just reflects the pattern in CO2. We have added citations of other papers that similarly present CH4 and CO2 budget comparisons using CO2-equivalents.

RE-REVIEW COMMENT:

My point was not to question the numerical value of the GWP factor. Rather, the point was to highlight the fact that, no matter how widely it is used (I have done it myself…), the CO2-eq concept is poorly applicable to natural ecosystems as their continuous GHG exchanges do not induce radiative forcing or a warming/cooling effect. However, this is how the CO2-eq flux is commonly interpreted, especially when calculated with fluxes of opposite direction. Moreover, I do not understand how "the disparity in the mass of carbon in CO2 and CH4 emissions …" explains the use of CO2-eq. Misleading in what sense? If we are addressing the carbon cycle, then it is most logical to calculate the sum as CO2-C + CH4-C, but I would not convert CO2-eq to C and call the result carbon budget.

We respectfully disagree on the first point. These natural systems do induce a net warming effect, as human-caused climate change thaws permafrost, changes hydrology, and leads to other indirect effects resulting in the Arctic becoming a net source of carbon instead of the historical sink it has been. Discussing the sum of CO2 and CH4 as a mass of C is misleading because the mass of C from CH4 emissions is often orders of magnitude less than that of CO2 emissions, and thus when examining trends CH4-C + CO2-C ~ CO2-C. We have updated the main text to specify "CO2-eq carbon budgets" in place of "carbon budgets" whenever we are referring to joint CO2 and CH4 scaled budget results.

Example see Lines 420: We can combine the posterior distributions of scaled carbon from all three footprint model results to calculate a single CO2-eq carbon budget estimate that accounts for across-model uncertainties.

(4)

REVIEW COMMENT - Table 1: It would be useful to show similar percentages for the average footprint-weighted land cover proportions during the study period, estimated with different footprint models.

AUTHORS' RESPONSE - The full distributions for the footprint-weighted landcover proportions for each of the three footprint models are in Figure 2 and in the SI. We chose to display the tower area proportions instead of footprint-weighted averages since none of the distributions are Gaussian and displaying the average is not a useful metric or central-tendency in

this case. We believe that including the actual distributions where the range and variance are clear is a better way to communicate this.

RE-REVIEW COMMENT:

Consider a flux (or any surface parameter) q_i that is constant over a certain period of time; i denotes the land cover class. It is easy to show that the mean flux observed by EC during this period can be expressed as sum_i(<f_i> q_i), where f_i is the 30-min footprint-weighted areal proportion of the land cover class i and <.> denotes temporal averaging. Thus, <f_i> indicates the relative contribution of each land cover class to the mean GHG flux or balance during the averaging period. I find this a useful metric that does not depend on the distribution of the 30-min f_i data.

That is an interesting comment and we will consider it in future work.
(5)

REVIEW COMMENT - line 127: What criterion was used for nonstationarity?

AUTHORS' RESPONSE - We used the Foken et al. 2004 method, as is cited in line 126 (implemented in EddyPro). This method assigns integers to indicate fluxes that pass or fail these QA/QC checks on the high frequency data. This method is a widely used in established EC networks (e.g. Ameriflux and Fluxnet).

RE-REVIEW COMMENT:

Yes, but the question was about the value adopted for the relative non-stationarity parameter. If you wish to refer to the QA/QC flagging system, then please specify in the manuscript how this was applied.

We have clarified that we removed bad data flags (value =2) from the overall flagging described in the updated citation.

*Lines 125-126: We removed fluxes with nonstationarity (flags = 2 in the overall flag system) (Mauder and Foken 2015).*

(6)

REVIEW COMMENT - line 395: Why would increasing the number of model parameters increase uncertainty? I would have assumed that this makes the model more flexible and thus decreases uncertainty.

AUTHORS' RESPONSE - Increasing uncertainty with increasing complexity is a common occurrence across modeling. See Puy et al. 2022 (https://www.science.org/doi/10.1126/sciadv.abn9450) for a nice summary.

RE-REVIEW COMMENT:

The Puy et al. (2022) paper deals with "process-based mathematical models that do not (or cannot) rely on a training and/or validation dataset". You do not use a process-based model and do have a training/validation dataset. While increasing uncertainty with increasing complexity may be common, it is not universal, hence my original comment.

In our study, because we are carrying through all of the uncertainty associated with every parameter into the regional carbon budgets, when we estimate twice as many parameters in more complex models we have more sources contributing their uncertainties. This section is explicitly about uncertainty in scaled carbon budgets, a derived quantity, not uncertainty around predicted values in the validation dataset, which is indeed often lower in the complex model versions. We have added text to clarify.

*Lines 399-400: For most months the complex map solutions were slightly more uncertain, a consequence of estimating almost twice as many parameters and carrying through all of their uncertainties.*